

# Selecting a conceptual hydrological model using Bayes' factors computed with Replica Exchange Hamiltonian Monte Carlo and Thermodynamic Integration

Damian N. Mingo[1], Remko Nijzink[2], Christophe Ley[3], and Jack S. Hale[1]

[1]Institute of Computational Engineering, Department of Engineering, Faculty of Science, Technology and Medicine, University of Luxembourg, Luxembourg.

[2]Environmental Research and Innovation, Luxembourg Institute of Science and Technology, Luxembourg.

[3]Department of Mathematics, Faculty of Science, Technology and Medicine, University of Luxembourg, Luxembourg.

**Correspondence:** Jack S. Hale (jack.hale@uni.lu)

**Abstract.** We develop a method for computing Bayes' factors of conceptual rainfall-runoff models based on thermodynamic integration, gradient-based replica-exchange Markov Chain Monte Carlo algorithms and modern differentiable programming languages. We apply our approach to the problem of choosing from a set of conceptual bucket-type models with increasing dynamical complexity calibrated against both synthetically generated and real runoff data from Magela Creek, Australia. We

show that using the proposed methodology the Bayes factor can be used to select a parsimonious model and can be computed robustly in a few hours on modern computing hardware. We introduce formal posterior predictive checks for the selected model. The prior calibrated posterior predictive p-value, which also tests for prior data conflict, is used for the posterior predictive checks. Prior data conflict is when the prior favours parameter values that are less likely given the data.

## 1    Introduction

Hydrologists are often faced with assessing the performance of models that differ in their complexity and ability to reproduce observed data. The Bayes factor (BF) is one method for selecting between models from an *a priori* chosen set (Berger and Pericchi, 1996). The appeal of the BF lies in its ability to implicitly and automatically balance model complexity and goodness-of-fit under few simplifying assumptions. The BF is also invariant to data and parameter transformations unlike information theory-based criteria such as Akaike information criteria (AIC) and Bayesian information criterion (BIC) (O'Hagan, 1997).

For example, a logarithmic transformation of the discharge or the square root of a parameter such as the flow rate can accelerate the convergence of the model, but it will not affect the computed BF.

However, the BF requires the computation of the marginal likelihood (the denominator in Bayes' theorem) for each model, which is a difficult and expensive integration problem. This expense and difficulty can be attributed to three main factors; the necessity of many model runs at different points in the parametric space; the possibly multi-modal and highly correlated



nature of the posterior that can lead to isolated and/or slowly mixing chains; and finally the inherent difficulty of the marginal likelihood integration problem.

Because of these difficulties, it is the case today that the BF is not widely used by practitioners, despite it being a crucial component in Bayesian model comparison, selection and averaging (Höge et al., 2019). This stands in contrast with the widely studied and used Bayesian parameter estimation procedure (Gelman et al., 2020). Consequently, model uncertainty is often

ignored, or assessed by either *ad hoc* techniques or information theoretic criteria (Birgé and Massart, 2007; Bai et al., 1999) that explicitly (rather than implicitly) penalise model complexity based on some measure of the number of parameters and under limiting assumptions, see e.g. Berger et al. (2001) for a full discussion.

## 1.1 Contribution

The overall contribution of this paper is to describe the development of a method, Replica exchange preconditioned Hamilto-

nian Monte Carlo (REpHMC), which, when used in conjunction with thermodynamic integration (TI), can be used to estimate the BF of competing conceptual rainfall-runoff hydrological models. Our approach for estimating the marginal likelihood combines TI for marginal likelihood estimation, Replica exchange Monte Carlo (REMC) for power posterior ensemble simulation and preconditioned Hamiltonian Monte Carlo (pHMC) for high-efficient gradient-based sampling which in sum we call the REpHMC + TI estimator. We demonstrate that REpHMC can sample from moderate-dimensional, strongly correlated and/or

multimodal distributions that frequently arise from hydrological models. In addition, REpHMC + TI can obtain posterior parameter estimates and the marginal likelihood simultaneously. We remark that Brunetti et al. (2019) also suggested, but did not explore, the idea of using REMC (therein called parallel tempering Monte Carlo) to improve chain mixing in hydrological models.

Another key contribution of our work compared with e.g. Brunetti et al. (2017, 2019) is the incorporation of recent ideas from

probabilistic programming for the automatic specification of the Bayesian inference problems (parameter and BF estimation). Utilising recent techniques from the literature on Neural Ordinary differential equations (ODEs) (Chen et al., 2018; Rackauckas et al., 2020; Kelly et al., 2020), we formulate a set of Hydrologiska Byråns Vattenbalansavdelning (HBV)-like models with extensible model complexity as a system of ODEs. Working in this framework allows us to use efficient high-order timestepping schemes for the numerical solution of the ODE system and to automatically derive the associated continuous adjoint ODE

system. With this adjoint system we can efficiently calculate the derivative of the posterior functional with respect to the model parameters, a necessary step for working with gradient-based samplers such as Hamiltonian Monte Carlo (HMC). We emphasise at this point that our approach is largely free of manual tuning parameters and straightforward to implement in a differentiable programming framework (we use TensorFlow probability (TFP) with the JAX backend, but the ideas are applicable in similar frameworks such as Stan or PyMC3). We remark that a recent more theory-focused paper (Henderson

and Goggans, 2019) also proposed using TI with HMC via the Stan probabilistic programming language, but with results for non-time series models and without using REMC, which is an important aspect of our approach.

After model selection via the BF, it is essential to check if the chosen model can generate the observed data. Hydrographs show the time series of stream flow. However, formal goodness-of-fit testing is necessary since it is challenging to see a



mismatch in hydrographs for dense data. We therefore use the prior calibrated posterior predictive p-value (PCPPP), which
simultaneously tests for prior data conflict and discrepancies in the model for further improvements.

In summary, this paper is the first to propose the REpHMC + TI method in a probabilistic programming framework for the
estimation of marginal likelihoods related to hydrological systems in view of model selection. We demonstrate the performance
of our method by showing a) a validation of the methodology using an analytically tractable model, b) its improved efficiency
with respect to classical methods using artificially generated data, and c) an application of a Bayes factor based model selection
on real rainfall/runoff data collected from the Magela Creek catchment in Australia.

Our overall perspective is that these techniques have the potential to bring robust model comparison techniques based on
BF closer to everyday hydrological modelling practice. Aside from the algorithmic developments in this paper, a necessary
technological requirement would be the (re-)implementation of hydrological models in a differentiable programming language,
e.g. JAX, PyTorch or TensorFlow, rather than in a traditional language such as C, Fortran or Python. While using modern
differentiable programming techniques is commonplace for model developers working with machine-learning type approaches,
e.g. neural networks, it is less commonly used, but no less applicable, for more traditional hydrological modelling approaches
like the Ordinary differential equation (ODE)-based HBV-like system we consider here. We hope our results encourage more
hydrologists to consider differentiable programming tools for conceptual model implementation given the advantages that
derivative-based sampling and optimisation algorithms bring to the table in terms of computational efficiency and improved
insight, e.g. model selection.

## 1.2   Background

Looking outside of hydrology, there are a number of notable works that have developed techniques for numerically estimating
the BF. A recent comprehensive review by Llorente et al. (2023) discusses the relative advantages of commonly used methods
for computing the marginal likelihood, and consequently, the BF, such as naive Monte Carlo methods, harmonic mean estimator
(Newton and Raftery, 1994), generalised harmonic mean estimator (Gelfand and Dey, 1994), importance sampling and Chib's
method (Chib and Jeliazkov, 2001; Chib, 1995), bridge sampling (Meng and Wong, 1996; Gelman and Meng, 1998), nested
sampling (Skilling, 2004, 2006) and finally thermodynamic integration (Calderhead and Girolami, 2009; Lartillot and Philippe,
2006; Ogata, 1989), the technique that we choose to use in this study. Thermodynamic integration is well suited for high
dimensional integrals (Ogata, 1989, 1990), and physics-based models. The naive Monte Carlo is unstable and usually not
efficient, requiring a huge number of samples for convergence. The importance sampling and harmonic estimators require a
suitable choice of the importance and proposal distributions, respectively. The performance of bridge sampling also depends
on a good choice of proposal distribution, which in practice is not straightforward *a priori*. The main difficulty with nested
sampling is generating samples from a truncated prior as the threshold increases (Chopin and Robert, 2010; Henderson and
Goggans, 2019). However, the efficiency of Chib's method depends on how close an arbitrary value is to the posterior mode
(Dai and Liu, 2022). Hug et al. (2016) illustrated that Chib's method significantly underestimates the marginal likelihood of a
bimodal Gaussian mixture model.



Turning our attention to works within hydrology that develop methods for computing Bayes factors, to the best of our knowledge, the seminal work by Marshall et al. (2005) was the first to propose computing Bayes factors for hydrological model selection. Marshall et al. (2005) used Chib's method to estimate the marginal likelihood of conceptual models. More recently various other authors (Liu et al., 2016; Brunetti et al., 2019, 2017; Volpi et al., 2017; Cao et al., 2019; Brunetti and Linde, 2018; Marshall et al., 2005) have considered the computation of Bayes factors in a hydrological or hydrogeological context.

Perhaps most closely related to our study are the recent works of Brunetti et al. (2019, 2017); Brunetti and Linde (2018) who computed Bayes factors for conceptual hydrogeological models with thermodynamic integration techniques. Brunetti et al. (2017) compared naive Monte Carlo, bridge sampling based on the proposal distribution developed by Volpi et al. (2017), and the Laplace metropolis method in terms of calculating the marginal likelihood of conceptual models. Like most studies, the naive Monte Carlo approach performed poorly. Also, Brunetti and Linde (2018) computed the marginal likelihood using methods based on a proposal distribution, notably bridge sampling. Several marginal likelihood estimation methods have been compared within hydrological studies. For example, Liu et al. (2016) found that thermodynamic integration gives consistent results compared to nested sampling and is less biased.

Many studies in hydrology, e.g. Zhang et al. (2020); Brunetti et al. (2017); Zheng and Han (2016); Shafii et al. (2014); Laloy and Vrugt (2012) and Kavetski and Clark (2011) have used the differential evolution adaptive Metropolis (DREAM) algorithm (Vrugt, 2016) for posterior parameter inference. Volpi et al. (2017) introduced a method to construct the proposal distribution for bridge sampling and integrated it with the DREAM algorithm. However, it still requires the user to choose the number of Gaussian distributions for the Gaussian mixture proposal distribution. The DREAM algorithm has been developed with an acceptance rate similar to the random walk Metropolis (RWM) algorithm, which has an optimal acceptance rate of 0.234 (Gelman et al., 1996b; Roberts and Rosenthal, 2009). The acceptance rate or probability is the proportion of the proposed samples accepted in the Metropolis-Hastings algorithm. DREAM has an optimal acceptance probability of around 0.23 for problems with dimensions greater than five (Vrugt et al., 2008). In contrast, a gradient-based sampler such as HMC, which we use in this work, typically has a far higher optimal acceptance rate of around 0.65 (Radford M. Neal, 2011; Beskos et al., 2013). In addition, gradient-based samplers show improved chain mixing properties in high dimensions and on posteriors with strongly correlated parameters (Radford M. Neal, 2011). The studies by Liu et al. (2016) and Brunetti et al. (2017, 2019) that use the BF use posterior samples from the DREAM algorithm, which has a lower acceptance ratio than the HMC method. In addition, because gradient-based samplers incorporate information about the local geometry of the posterior, they are usually easier to tune to achieve the optimal acceptance rate, particularly in the moderate or high-dimensional parameter setting (num. parameters > 5). That is, to achieve the optimal acceptance rate, one may need to select a value for a parameter of the algorithm, as described in Algorithm 3.

The rest of the paper is organized as follows. Section 2 is about conceptual hydrological models and Bayesian methodology, which includes model formulation, prior and likelihood construction, posterior predictive checks, numerical methods, and algorithms. Section 3 contains the results and discussions, while the conclusions are provided in Section 4. There is also a list of acronyms at the end.



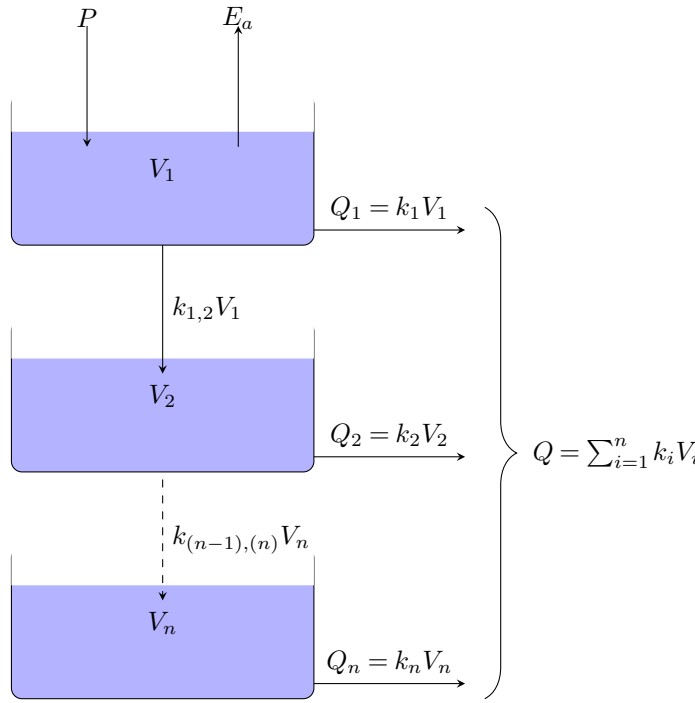

**Figure 1.** Schematic representation of HBV-like ODE model with $n$-buckets according to the notations in the text. The blue boxes represent the buckets with given state $V_1$ to $V_n$. The solid arrows represent mass flows between buckets, into the system or out of the system. The dashed arrow represents the collective mass flow between multiple buckets.

## 2 Methodology

This section describes the model formulation, including prior specification, likelihood construction, algorithms used, and implementation in differentiable software. We leave other modelling aspects, like the type of priors used, for the next section, where we present experiments.

### 2.1 Conceptual models

We develop a set of rainfall-runoff conceptual hydrological models in the framework of continuous dynamical systems that can be written as a system of ODEs of the following form

$$
\begin{aligned}
V_t &= f(t, V, \theta) \quad \forall t \in (0, \bar{T}], \\
V(t = 0) &= \hat{V},
\end{aligned}
\tag{1}
$$

where $V$ are the $n$ system states, $V_t := \frac{dV}{dt}$ is the derivative of the state with respect to the time variable $t$, $\bar{T}$ is the final time, $\hat{V} \in \mathbb{R}^n$ are the initial conditions, $f$ are known functions, and $\theta \in \mathbb{R}^p$ is a vector containing the $p$ model parameters.



For the purpose of the results in this paper, we derive a set of HBV-like models under the principle of conversation of mass. The algorithms developed in this study can be applied to other bucket-type models, e.g. Parajka et al. (2007); Jansen et al. (2021) or those described in the comprehensive MARRMoT rainfall-runoff models toolbox (Trotter et al., 2022). In comparison with the 'standard' HBV model (Bergström, 1976), our model lacks snow and a routing routine and we choose to replace the traditional soil moisture routine with a linear reservoir. A schematic representation of mass flow between the buckets system is given in Fig. 1. The system states $\{V_1, \ldots, V_n\}$ $[L^3]$, where $L$ is a generic length unit, represent the volume of water in the $i$-th bucket and $n$ is the total number of buckets. The system of ODEs for general $n \geq 1$ can be written

$$(V_1)_t = P - E_a - k_1 V_1, \quad n = 1, \tag{2a}$$

$$(V_1)_t = P - E_a - k_1 V_1 - k_{1,2} V_1, \quad n \geq 2 \tag{2b}$$

$$(V_i)_t = k_{(i-1),(i)} V_{i-1} - k_i V_i - k_{(i),(i+1)} V_i, \quad i = 2, \ldots, n-1, \quad n \geq 3, \tag{2c}$$

$$(V_n)_t = k_{(n-1),(n)} V_{n-1} - k_n V_n, \quad n \geq 2, \tag{2d}$$

$$V(t=0) = \hat{V}, \tag{2e}$$

$$E_a = \frac{E_p}{V_{\max}} V_1, \tag{2f}$$

$$Q = \sum_{i=1}^{n} k_i V_i. \tag{2g}$$

The parameters $k_{(i-1),(i)}$ $[T^{-1}]$, $i = 2, \ldots, n$, are the interbucket recession coefficients, where $T$ is a generic time unit. The parameters $k_{(i)}$ $[T^{-1}]$, $i = 1, \ldots, n$, are the outflow recession coefficients. The total outflow $Q$ $[L^3 T^{-1}]$ specified in Eq. (2g) is the noiseless quantity $y$ used in the upcoming calibration and model selection procedures. The precipitation $P$ $[L^3 T^{-1}]$ is an *a priori* known function of time. Potential evaporation $E_p$ $[L^3 T^{-1}]$ is a known function of time, whereas actual evaporation $E_a$ $[L^3 T^{-1}]$ is a function of $E_p$, and $V_{\max}$ $[L^3]$ through Eq. (2f), where $V_{\max}$ is the maximum amount of water the soil can store. We remark that the term $E_p / V_{\max}$ in Eq. (2f) has units $[L^3 T^{-1}]$ and can therefore be thought of as a dynamic recession coefficient with the dynamic behaviour controlled by the known time-varying potential evapotranspiration function $E_p$.

The parameter vector $\theta \in \mathbb{R}^p$ associated with the model is then

$$\theta := \{\underbrace{V_{\max}}_{1}, \underbrace{k_1, \ldots, k_n}_{n}, \underbrace{k_{2,1}, \ldots, k_{(n-1),(n)}}_{n-1}, \underbrace{\hat{V}_1, \ldots, \hat{V}_n}_{n}\} \tag{3}$$

The number of buckets can be varied by adjusting $n \in \mathbb{N}^+$, leading to a set of models $\{M_1, \ldots, M_n\}$ each with $n$ states and $p = 3n$ parameters.



## 2.2 Bayesian methodology

We briefly restate the Bayes theorem in order to set our notation. If $y$ is the data and $\theta$ the parameter vector associated with a model $M$, then Bayes' theorem in Eq. (4) defines the posterior probability of $\theta$ as

$$\underbrace{\pi(\theta|y,M)}_{\text{posterior}} = \frac{\overbrace{f(y|\theta,M)}^{\text{likelihood}}\overbrace{\pi(\theta|M)}^{\text{prior}}}{\underbrace{p(y|M)}_{\text{marginal (averaged) likelihood}}} = \frac{f(y|\theta,M)\pi(\theta|M)}{\int f(y|\theta,M)\pi(\theta|M)d\theta}. \tag{4}$$

The prior is a probability distribution of a parameter before data is considered. It can incorporate expert knowledge, historical results or any belief about the model parameters. The likelihood tells us how likely various parameter values could have generated the observed data. The denominator in Bayes' theorem

$$p(y|M) = \int \overbrace{f(y|\theta,M)}^{\text{likelihood}}\overbrace{\pi(\theta|M)}^{\text{prior}}\,\mathrm{d}\theta, \tag{5}$$

is called the marginal likelihood. The marginal likelihood tells us how likely the model supports the data. The distribution of the parameters given the data is known as the posterior and is proportional to the product of the likelihood and the prior. In the Bayesian paradigm, all inference is based on the posterior.

### 2.2.1 Likelihood construction

In this section, we drop the explicit index on the model for notational convenience. We define a solution operator $G_{\text{sol}} :$
$\mathbb{R}^{3n} \to X$ that maps a parameter vector $\theta_j$ to the total outflow function $Q$. Concretely, this solution operator is calculated by numerically solving Eqs. (2a) to (2g). We then define the observation operator $G_{\text{obs}} : X \to \mathbb{R}^p$ which evaluates the solution $Q \in X$ at a set of $p$ points in time $\{t_1,\ldots,t_p\}$.

We assume the following standard Gaussian white noise model for the observed data: $y = G_{\text{obs}}G_{\text{sol}}(\theta) + \eta$ where $\eta \sim$ $\text{MVN}(0,\sigma^2 I_p)$ with MVN a multivariate normal distribution with mean $0 \in \mathbb{R}^p$ and covariance $\sigma^2 I_p \in \mathbb{R}^{p\times p}$, with $\sigma^2 \in \mathbb{R}$ the
175 variance of the measurement noise and $I_p$ the usual $p$-dimensional identity matrix. Let $G := G_{\text{obs}}G_{\text{sol}} : \mathbb{R}^q \to \mathbb{R}^p$. By standard arguments it can be shown that $y|\theta \sim \text{MVN}(G(\theta),\sigma^2 I_p)$ resulting in the likelihood $f(y|\theta,M)$ in Eq. (4) being fully defined. For brevity, we leave precise prior specifications to the results in Section 3.

### 2.2.2 Model comparison

The marginal likelihood is also called the normalizing constant (Chen et al., 2000; Gelman and Meng, 1998), prior predictive
density, evidence (MacKay, 2003) or integrated likelihood (Lenk and DeSarbo, 2000; Gneiting and Raftery, 2007). This quantity is essential to the definition of the Bayes factor. Indeed, the Bayes factor for two competing models, $M_1$ and $M_2$ is the




ratio of their marginal likelihoods

$$\mathrm{BF}_{12} = \frac{p(y|M_1)}{p(y|M_2)} = \frac{\int f(y|\theta_1, M_1)\pi(\theta_1|M_1)d\theta_1}{\int f(y|\theta_2, M_2)\pi(\theta_2|M_2)d\theta_2}. \tag{6}$$

Since BF is a ratio, a value greater than one means that $M_1$ should be preferred to $M_2$, and vice-versa for a value smaller than one. Kass and Raftery (1995) proposed a measure of the strength of evidence (Table 1) that we will use throughout this paper to interpret the Bayes factors.

An appealing feature of the BF is its consistency in the limit of a high number of observations. Proofs of consistency for non-nested models are in Casella et al. (2009). For other cases, including nonparametric models, a review and detailed study of consistency can be found in Chib and Kuffner (2016). Also, information theoretic model selection approaches usually require an explicit penalty for the number of model parameters (model complexity). In contrast, the BF implicitly penalises the complexity of the model. That is we do not need to assign a penalty for model complexity since it is already accounted for in the marginal likelihood and hence the BF.

**Table 1.** Interpretation of the Bayes factor (Kass and Raftery, 1995)

| $\log_{10} \mathrm{BF}_{12}$ | $\mathrm{BF}_{12}$ | Evidence in favour of model 1 |
|---|---|---|
| 0 to 1/2 | 1 to 3.2 | Not worth more than a bare mention |
| 1/2 to 1 | 3.2 to 10 | Substantial |
| 1 to 2 | 10 to 100 | Strong |
| >2 | >100 | Decisive |

### 2.2.3 Posterior predictive checks

Model selection does not reveal discrepancies between the predictions from the chosen model and observed data. Hence posterior predictive checks (PPCs) are also necessary to see if the selected model can replicate the observed data (Gelman et al., 1996a). PPCs can be graphical or formal. Graphical PPCs consist in making plots of predictions from the chosen model and the observed data to reveal discrepancies. Formal PPC entails calculating a posterior predictive p-value (PPP). The concept of posterior predictive checking was introduced by Rubin (1984) and later generalised by Gelman et al. (1996a) under the name PPP where a discrepancy measure can depend on the model parameters. PPCs are the Bayesian equivalent of frequentist goodness-of-fit tests, with the difference that the PPP can be based on any discrepancy measure, not just a statistic. To compute the PPP, the chosen discrepancy measure is calculated based on replicated data $y^{\mathrm{rep}}$, drawn from the predictive distribution $\pi(y^{\mathrm{rep}}|y_{\mathrm{obs}}) = \int f(y^{\mathrm{rep}}|\theta)\pi(\theta|y_{\mathrm{obs}})d\theta$, and compared with that based on observed data. In mathematical terms, we wish to approximate the theoretical probability

$$\mathrm{ppp}(y_{\mathrm{obs}}) = Pr\big[D(y^{(\mathrm{rep})}, \theta) \geq D(y_{\mathrm{obs}}, \theta)|y_{\mathrm{obs}}\big]. \tag{7}$$





This quantity can be estimated as

$$\text{ppp}(y_{\text{obs}}) = \frac{1}{B} \sum_{i=1}^{B} I\big[D(y_i^{\text{rep}}, \theta_i) \geq D(y_{\text{obs}}, \theta_i)\big] \qquad (8)$$

where $I[A]$ stands for the indicator function which takes the value 1 if $A$ occurs and 0 otherwise, $y_{obs}$ is the observed dataset, $y_i^{\text{rep}}$ is a replicated dataset from the posterior predictive distribution, $B$ is the number of replicated datasets, while $\theta_i$ is a single draw from the posterior distribution.

Unlike the frequentist p-value, the interpretation of the PPP is not straightforward since it does not follow a uniform distribution but is concentrated around 0.5 (Meng, 1994). When the p-value has a uniform distribution, the type I error can be controlled. For the frequentist p-value, the probability of falsely rejecting a null hypothesis, which is referred to as a type I error rate, can be set to a fixed value. Typically, this value is prespecified at 0.05 or 0.01. On the contrary, it is difficult to fix the type I error rate for the PPP. Hence, we might fail to reject poor models for a given PPP at a chosen type one error (Gelman, 2013;

Hjort et al., 2006). For this reason, we computed the prior calibrated posterior predictive p-value (PCPPP) introduced by Hjort et al. (2006) that has a uniform distribution and the same interpretation as a classical p-value. For more on the Type I error and the distribution of the p-value, refer to Hung et al. (1997) and for Bayesian p-values, see Zhang (2014). To calculate the PCPPP, a PPP based on data from the prior predictive distribution $\pi(y_{\text{prior}}) = \int f(y^{\text{rep}}|\theta)\pi(\theta)d\theta$ is computed and compared with a PPP based on replicated data from the posterior predictive distribution:

$$\text{pcppp}(y_{\text{obs}}) = \frac{1}{B} \sum_{i=1}^{B} I\big[\text{ppp}(y_{\text{prior}_i}^{\text{rep}}) \leq \text{ppp}(y_{\text{obs}})\big],$$

where $\text{ppp}(y_{obs})$ is obtained by Eq. (8) and $\text{ppp}(y_{\text{prior}_i}^{\text{rep}})$ can be in a similar way. From this equation, it becomes visible that the PCPPP can also reveal prior data conflicts. A PCPPP greater than a prespecified type I error, say 0.05, means that the prior distribution and model support the data at the level 0.05. The PPP can as well be calibrated based on posterior samples (Hjort et al., 2006; Wang and Xu, 2021).

**2.3 Numerical methods**

In this section we discuss the proposed new numerical method Replica exchange Hamiltonian Monte Carlo (REHMC) + TI that we employ to simultaneously draw posterior samples and estimate the marginal likelihood. We recommend the reader refer to Fig. 2 and its caption for a high-level overview of the approach before continuing to the detailed descriptions below.

**2.3.1 Thermodynamic integration**

Thermodynamic integration (TI) has its origins in theoretical physics, where it is used to calculate free energy differences between systems (Torrie and Valleau, 1977) before appearing in the statistical literature as path sampling (Gelman and Meng,





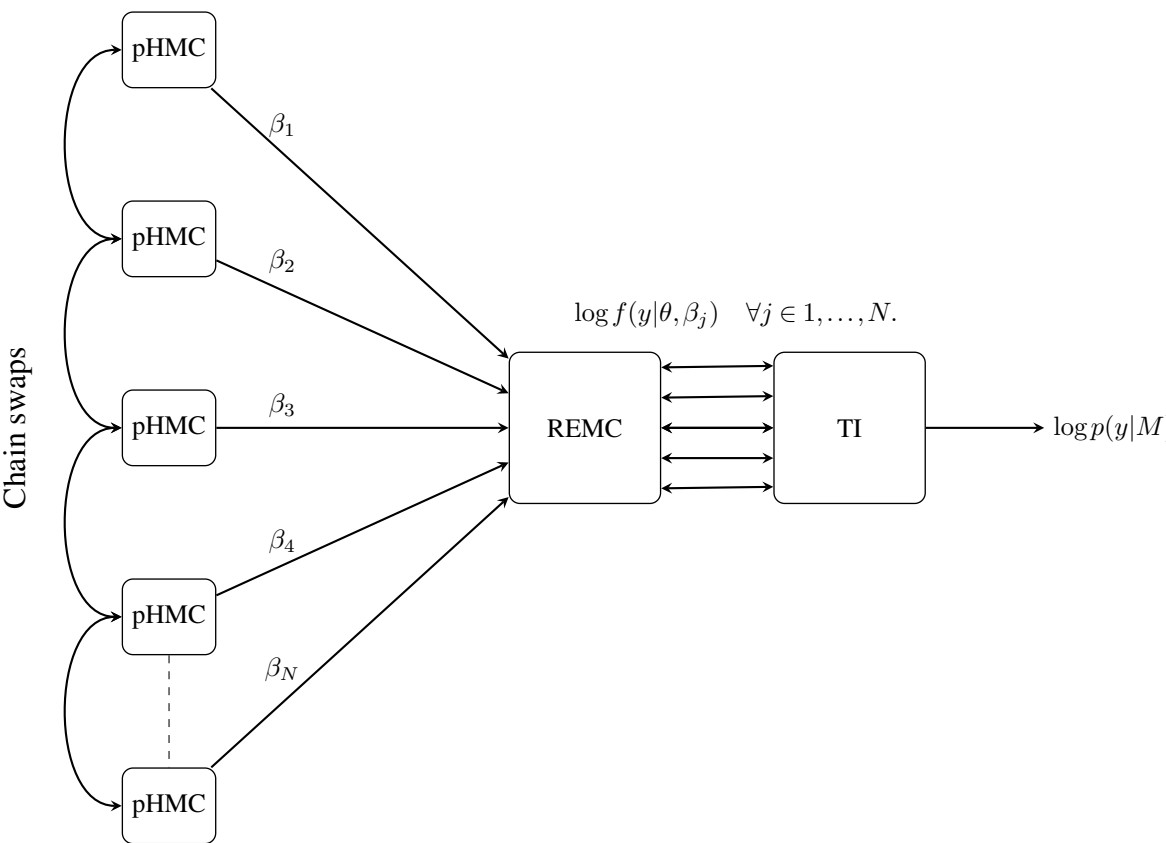

**Figure 2.** Overall schematic of the REpHMC+TI algorithm for estimating the marginal likelihood for a given model $M$. Working from left to right, $N$ pHMC samplers are run at different values of the inverse temperature parameter $\{\beta_1, \beta_2, \ldots, \beta_N\}$ with $0 \leq \beta_j \leq 1, j = 1, \ldots, N$, to simulate from the power posterior $\log f(y; \theta_i, \beta_j)$. The REMC algorithm is responsible for swapping the state between adjacent chains according to the Metropolis-Hastings criteria. Finally, the TI methodology is used to calculate an estimate of the marginal likelihood $\log p(y|M)$. Note that in terms of setup, information flows from right to left, i.e. the discretisation of the TI integral is responsible for setting the number $N$ and values of inverse temperatures $\beta_1, \ldots, \beta_N$.

1998), a method for calculating marginal likelihoods. TI converts a high-dimensional integral into a one-dimensional integration problem over a unit interval.

To derive the TI estimate of the marginal likelihood $p(y)$, we first raise the likelihood to the power $0 \leq \beta \leq 1$ to form the

power posterior (Friel and Pettitt, 2008)

$$\pi_{\text{power}}(\theta|y,\beta) = \frac{\left[f(y|\theta)\right]^{\beta} \pi(\theta)}{p(y|\beta)}, \tag{9}$$

with

$$p(y|\beta) = \int \left[f(y|\theta)\right]^{\beta} \pi(\theta) \, \mathrm{d}\theta. \tag{10}$$





When $\beta = 0$, the power posterior is the same as the prior distribution. When $\beta = 1$, we have the standard posterior distribution.
This makes a continuous path between the prior and the posterior distributions.

Taking the logarithm on both sides of Eq. (10) and using the chain rule, a differentiation with respect to $\beta$ yields

$$
\begin{aligned}
\frac{\partial}{\partial \beta} \log p(y|\beta) &= \frac{1}{p(y|\beta)} \frac{\partial}{\partial \beta} p(y|\beta) \\
&= \frac{1}{p(y|\beta)} \int \frac{\partial}{\partial \beta} \left[ f(y|\theta) \right]^{\beta} \pi(\theta) \, \mathrm{d}\theta \\
&= \frac{1}{p(y|\beta)} \int \left[ f(y|\theta) \right]^{\beta} \log f(y|\theta) \pi(\theta) \, d\theta \\
&= \int \frac{\left[ f(y|\theta) \right]^{\beta} \pi(\theta)}{p(y|\beta)} \log f(y|\theta) \, d\theta \\
&= \mathbb{E}_{p(\theta|y,\beta)} [\log f(y|\theta)],
\end{aligned}
\tag{11}
$$

where $\mathbb{E}_{p(\theta|y,\beta)}$ is the expectation with respect to the power posterior. Integrating both sides of equation (11) with respect to $\beta$
gives the log of the marginal likelihood of interest $p(y)$ in terms of an integral on $\beta$

$$
\log p(y) = \int_{0}^{1} \mathbb{E}_{p(\theta|y,\beta)}[\log f(y|\theta)] \, d\beta,
\tag{12}
$$

This manipulation allows us to find a way to approximate the value of $p(y)$. Computationally, posterior samples are drawn for
each value of $\beta$. The values are then evaluated in the log-likelihood, and the mean for each value of $\beta$ is obtained. The integral
(12) on $\beta$ can be estimated using the trapezoidal rule as follows:

$$
\log p(y) = \sum_{j=1}^{N} \frac{(\beta_j - \beta_{j-1})}{2} \left[ \mathbb{E}_{p(\theta|y,\beta_j)} \log f(y|\theta) + \mathbb{E}_{p(\theta|y,\beta_{j-1})} \log f(y|\theta) \right].
$$

The Monte Carlo estimate of the expectations can then be obtained by

$\displaystyle \log p(y) \approx \sum_{j=1}^{N} \frac{(\beta_j - \beta_{j-1})}{2} \left[ \frac{1}{S} \sum_{i=1}^{S} \log f(y|\theta_i, \beta_j) + \frac{1}{S} \sum_{i=1}^{S} \log f(y|\theta_i, \beta_{j-1}) \right],$   (13)

where $j = 1, \ldots, N$ is the index for the $\beta$ values and $S$ is the number of posterior samples for each $\beta$ value. The accuracy
of the TI estimate depends on the integration rule on $\beta$, i.e. the number of $\beta$ values and the spacing of the values, and the
convergence of the Markov Chain Monte Carlo (MCMC). The most commonly employed path is a geometric path (Calderhead
and Girolami, 2009)

$\displaystyle \beta_j = \left( \frac{j}{N} \right)^5, \quad j = 1, \ldots, N.$   (14)





The number of $\beta_j$ values can be adaptively chosen as a tradeoff between model convergence and computational efficiency, for instance, see Vousden et al. (2016). The complete TI algorithm is presented in Algorithm 1.

---

**Algorithm 1** Thermodynamic integration (TI)

---

**Input:** $\beta \{\beta = \{1, \ldots, 0\}$ : schedule of inverse temperatures based on trapezoidal rule of size $N$, $S$ is the number of samples per replica.$\}$
**Output:** Log marginal likelihood ($\log p(y)$).
 1: REpHMC($\beta, S$) {Run the REpHMC algorithm, see section 2.3.2.}
 2: Estimate $\log p(y)$ by the definition of the quadrature rule, e.g. trapezoidal rule

$$\log p(y) \approx \sum_{j=1}^{N} \frac{(\beta_j - \beta_{j-1})}{2} \left[ \frac{1}{S} \sum_{i=1}^{S} \log f(y|\theta_i, \beta_j) + \frac{1}{S} \sum_{i=1}^{S} \log f(y|\theta_i, \beta_{j-1}) \right].$$

---

### 2.3.2 Replica exchange Monte Carlo

The REMC algorithm was introduced by Swendsen and Wang (1986). Geyer (1991) presented a similar formulation to the statistical community under the name Metropolis-coupled MCMC. REMC is a generic algorithm in that it can be combined with other algorithms. Miasojedow et al. (2013) combined REMC with random walk Metropolis (RWM). RWM is a gradient-free algorithm in that it generates posterior samples from the target distribution by randomly sampling from a proposal distribution. We combine REMC with HMC, which gives the new algorithm REHMC explained in the rest of this section. When REMC is combined with pHMC, we get the REpHMC. The REpHMC gives a higher effective sample size than REHMC. The effective sample size is the number of independent samples with the same amount of information as correlated samples. Each sample in a Markov chain is correlated to the preceding sample, so the samples have less information than independent samples. The effective sample size takes into account this autocorrelation. The main idea of REMC is that an ensemble of power posterior chains known as replicas run in parallel. The likelihood of these chains is raised to values from zero to one. These values are called inverse temperatures. Each replica performs a Metropolis update to get the next value at each iteration. The replica pairs are randomly selected, and an attempt is made to swap the current values of the replica pairs. A swap is accepted or rejected according to the Metropolis-Hastings algorithm. The swapping accelerates convergence to the target distribution, avoids chains becoming trapped in topologically isolated areas of the parameter space, and improves the mixing of the chains. REMC is also known as parallel tempering (Hansmann, 1997; Earl and Deem, 2005). When the method has an iterated importance sampling step, it is known as population Monte Carlo (PMC) (Iba, 2000; Cappé et al., 2004). However, the term PMC has also been used for methods without an importance sampling step (Calderhead and Girolami, 2009; Friel and Pettitt, 2008; Mingas and Bouganis, 2016).

In the context of this paper, the samples of the replica with $\beta = 1$ are used to estimate the posterior parameters, while the entire ensemble is used as input for TI to calculate the marginal likelihood. The REpHMC is summarised in Algorithm 2.





---

**Algorithm 2** Replica Exchange preconditioned Hamiltonian Monte Carlo (REpHMC)

---

**Input:** $S, L, \epsilon, \beta$ {$S$: number of samples, $L$: number of leapfrog steps, $\epsilon$: leapfrog stepsize, $\beta = \{1, \ldots, 0\}$ : inverse temperatures.}
**Output:** $(\theta_{\beta i}^{t+1}, \theta_{\beta j}^{t+1}, \theta_{\beta k}^{t+1}, \ldots, \theta_{\beta N}^{t+1})$ {Posterior samples for each $\beta$}.
1: **for all** $\beta$ **do**
2:  **for** $t = 1$ to $n$ **do**
3:   pHMC($S, L, \epsilon, \beta$){Run pHMC algorithm for $\beta$ values in parallel}.
4:   **function** RE($\theta_{\beta i}^{t+1}, \theta_{\beta j}^{t+1}, \ldots, \theta_{\beta N}^{t+1}$){Swap replicas}.
5:   Randomly select a pair of samples from two different chains.
6:   Attempt to swap samples. For example $(\theta_{\beta i}^{t+1}, \theta_{\beta j}^{t+1})$.
7:   Swap acceptance probability is

$$\alpha_{i,j} = \min \left( 1, \frac{\pi_i(\theta_{\beta j})\pi_j(\theta_{\beta i})}{\pi_i(\theta_{\beta i})\pi_j(\theta_{\beta j})} \right)$$

8:   where $\pi_i(\cdot)$ is the power posterior .
9:   $u \sim U(0,1)$
10:   **if** $u \le \alpha_{i,j}$ **then**
11:    $(\theta_{\beta i}^{t+1}, \theta_{\beta j}^{t+1}) \leftarrow (\theta_{\beta j}^{t+1}, \theta_{\beta i}^{t+1})$
12:   **else**
13:    $(\theta_{\beta i}^{t+1}, \theta_{\beta j}^{t+1}) \leftarrow (\theta_{\beta i}^{t+1}, \theta_{\beta j}^{t+1})$
14:   **end if**
15:   **return** $(\theta_{\beta i}^{t+1}, \theta_{\beta j}^{t+1}, \theta_{\beta k}^{t+1}, \ldots, \theta_{\beta N}^{t+1})$
16:  **end for**
17: **end for**

---

Like any sampling method, the REpHMC's convergence should be assessed. We used both trace plots and formal diagnostic tests to check for convergence of the Markov chain since there is no universal robust test for convergence (Cowles and Carlin, 1996). The most widely used method to assess the convergence of Markov chains is the potential scale reduction factor $\hat{R}$, developed by Gelman and Rubin (1992) and extended by Brooks and Gelman (1998). Recently, an improved factor $\hat{R}$ was proposed by Vehtari et al. (2021). For $\hat{R}$ to be a valid statistic, the chains must be independent of each other. In REHMC, the chains are not independent due to swapping. Therefore, we used methods that require one chain or replica per temperature, namely the Geweke diagnostic (Geweke, 1991) and the integrated autocorrelation time (IAT) (Geyer, 1992; Kendall et al., 2005). For the sake of brevity, we do not explain these concepts here but instead refer the reader to the respective papers.

### 2.3.3 Hamiltonian Monte Carlo

HMC is a gradient-based technique used to sample from a continuous probability density (Duane et al., 1987). HMC scales better in high dimensions than gradient-free samplers, such as nested sampling, due to the inclusion of derivative information (Ashton et al., 2022). Therefore, many applications combine HMC and gradient-free samplers. For example, Elsheikh et al. (2014) has combined HMC and nested sampling. HMC is based on the Hamiltonian, which describes a particle's position and momentum at any time. New positions are known by solving Hamilton's equations of motion for position and momentum. In



Bayesian inference, the Hamiltonian $H(\theta, \rho)$ in Eq. (15) describes the evolution of a $d$ dimensional vector ($\theta$) of parameters and a corresponding $d$ dimensional vector of auxiliary momentum ($\rho$) variables at any time, $t$.

$$
\begin{aligned}
H(\theta, \rho) &= -\log f(y|\theta)\pi(\theta) + \frac{1}{2}\rho^T m\rho \\
&= U(\theta) + K(\rho)
\end{aligned}
\tag{15}
$$


In Eq. (15), $m$ is the positive definite mass matrix because it is the inverse of the covariance matrix. $U(\theta)$ is the desired posterior known as potential energy, and $K(\rho)$ is the kinetic energy that is a function of momentum. In practice, $\rho$ is sampled from a Gaussian distribution. To sample from the Hamiltonian, we take the partial derivatives, which give Hamilton's equations of motion

$$
\frac{d\theta}{dt} = \frac{\partial H}{\partial \rho} = \frac{\partial K}{\partial \rho}
\tag{16a}
$$


$$
\frac{d\rho}{dt} = -\frac{\partial H}{\partial \theta} = -\frac{\partial U}{\partial \theta}
\tag{16b}
$$

We now have a system of ODEs (Eqs. (16a) to (16b)). The leapfrog method (Duane et al., 1987; Radford M. Neal, 2011) is used to solve the Eqs. (16a) to (16b) and propose new values for the parameters. The accuracy of the leapfrog method depends on the discretisation step $\epsilon$.

Each HMC iteration consists of two steps (Radford M. Neal, 2011). In the first step, the momentum values for each parameter are sampled from a Gaussian distribution independent of the current $\theta$ values, $\rho_i \sim N(0, m)$ where $i = 1, \ldots, d$. Then using the current parameter and momentum values, $(\theta^t, \rho^t)$, the Hamiltonian is simulated using an appropriate time stepping method such as the leapfrog method (Betancourt, 2017). At the end of Hamiltonian dynamics, the momentum values are negated, and the new parameter values $(\theta^{t+1}, \rho^{t+1})$ are accepted or rejected using the Metropolis-Hastings criterion with acceptance

probability $\alpha$ where

$$
\alpha = \min\left[1, \exp\left(-U(\theta^{t+1}) + U(\theta^t) - K(\rho^{t+1}) + K(\rho^t)\right)\right].
\tag{17}
$$

The HMC is summarised in Algorithm 3. The mixing of the HMC chain depends on the number of leapfrog steps $L$ and the step size $\epsilon$. $L$ and $\epsilon$ can be automatically tuned during the warm-up phase of the algorithm (Hoffman and Gelman, 2014). The warm-up phase is the period during which posterior samples are discarded and is also called burn-in. In this work, $\epsilon$ was

automatically tuned by the dual averaging algorithm while $L$ was manually tuned. Dual averaging automatically adjusts $\epsilon$ during the warm-up of the HMC algorithm until a specific acceptance rate is achieved. We used an acceptance rate of 0.75, which is higher than the optimal acceptance rate of RWM based algorithms. This is the mean of various reported values and the default in TensorFlow probability.





## 2.4 Preconditioned Hamiltonian Monte Carlo

To increase the sampling efficiency of HMC, we have to reduce the correlation of the parameters, especially for ODE models. This is achieved by introducing a preconditioned matrix, $\mathbf{M}$ and hence the name pHMC. This leads to even faster convergence and higher effective sample sizes for each parameter (Girolami and Calderhead, 2011). In practice, the preconditioned matrix is the inverse of the covariance matrix of the target posterior. In contrast to HMC, where the momentum is sampled from a normal distribution, for pHMC, the momentum values are sampled from a multivariate Gaussian distribution with a covariance

matrix as the preconditioned matrix, $\rho \sim \mathrm{MVN}(0, \mathbf{M})$. The covariance matrix controls the correlation of the parameters. The rest of the algorithm for pHMC works as for HMC.

---

**Algorithm 3** Hamiltonian Monte Carlo (HMC), Notation following Radford M. Neal (2011)

---

**Input:** $S, L, \epsilon$ {$S$: number of samples, $L$: number of leapfrog steps, $\epsilon$: leapfrog stepsize.}
**Output:** $\theta^{t+1}$
  1: $\theta^0 \sim \pi(\theta)$ {Sample initial values from prior}
  2: **for** $t = 1$ to $n$ **do**
  3:     **if** $t = 1$ **then**
  4:        $\theta^t \leftarrow \theta^0$
  5:     **end if**
  6:     **for** $j = 1$ to $L$ **do**
  7:        $\rho_i^t \sim N(0, m)$ {Sample momentum values, where $i = 1 \ldots, d$ and $d$ is the dimension.}
  8:        $(\theta^*, \rho^*) \leftarrow \mathrm{Leapfrog}(\theta^t, \rho^t, \epsilon)$
  9:        $\rho^t \leftarrow -\rho^*$
10:        $\alpha = \min\left(1, \exp\left(-U(\theta^*) + U(\theta^t) - K(\rho^*) + K(\rho^t)\right)\right)$
11:        $u \sim U(0, 1)$
12:        **if** $u \leq \alpha$ **then**
13:           $\theta^{t+1} \leftarrow \theta^*$
14:        **else**
15:           $\theta^{t+1} \leftarrow \theta^t$
16:        **end if**
17:     **end for**
18: **end for**
19: **function** $\mathrm{Leapfrog}(\theta^t, \rho^t, \epsilon)$ {solves the equations to propose new values}
20:     $\rho^* \leftarrow \rho^t - (\epsilon/2)\frac{\partial U}{\partial \theta}(\theta^t)$
21:     $\theta^* \leftarrow \theta^t + \epsilon m^{-1}\rho^*$
22:     $\rho^* \leftarrow \rho^* + (\epsilon/2)\frac{\partial U}{\partial \theta}(\theta^*)$
23: **return** $(\theta^*, \rho^*)$

---

## 2.5 Implementation aspects

In this section, we outline some of the more non-standard aspects of implementing the proposed methodology in the probabilistic programming language (PPL) TFP. Probabilistic programming (PP) is a methodology for performing computational

statistical modelling in which all elements of the Bayesian joint posterior $\pi(\theta|y, M)$ are specified in a PPL. Popular PPLs include Stan (Carpenter et al., 2017), PyMC3 (Salvatier et al., 2016) and TFP (Dillon et al., 2017). Once specified in a PPL,





the subsequent Bayesian parameter inference problem can then be handled semi-automatically. We refer the reader to the Code and Data availability statement for the full implementation and simply remark that the joint posterior for our problem can be defined in around 70 lines of TFP/JAX code.

We choose to use TFP in this study. From our experience, TFP is the most flexible and extensible PPL in terms of allowing advanced model specification and the ability to break out of the high-level interface and perform low-level operations. However, this flexibility comes at the cost of a steeper learning curve, particularly TFP's complex batch and event shape semantics (Dillon et al., 2017). We note that despite TensorFlow in the name, TFP is backend-agnostic and can run on top of various differentiable programming languages. We choose to run TFP on top of JAX, instead of the default choice of TensorFlow. Anecdotally, our

experience is that TFP on JAX has better runtime performance and is more robust than TFP on TensorFlow, particularly when working with ODE-based models. We use JAX with the CPU backend and double precision floating point representation, although in principle the GPU backend could also be used. TFP already includes an implementation of the HMC and REMC algorithms, the output of which can be used with TI for computing the marginal likelihood.

JAX can automatically perform arbitrarily composable forward and backward mode automatic differentiation of nearly

arbitrary computer programs. This is used to automatically differentiate the TFP specification of the negative log posterior $U(\theta)$ with respect to the model parameters $\theta$ for use within the HMC algorithm. As this approach is now standard, we refer the reader to Margossian (2019) for a detailed review.

For the automatic differentiation of the ODE model, we use the continuous adjoint approach. This approach is also called continuous backpropagation in the Neural ODE literature, see e.g. Kelly et al. (2020) and Höge et al. (2022) for an application

in hydrology. We follow the presentation in (Kidger, 2021) where a new set of adjoint ODEs is from the original continuous ODE system. This adjoint system is then discretised (backwards in time) using the same ODE solver as for Eq. (1), an explicit adaptive Dormund-Prince ODE integrator that is already included in JAX. It is worth remarking that while the continuous adjoint system is still derived automatically within JAX, the result is distinctly different to backwards differentiation through the steps of the forward ODE solver at the programmatic level. For more details, we refer the reader to Kidger (2021) for a

discussion of the different methods for automatically differentiating ODE systems and their relative tradeoffs.

Let $V$ be the solution to Eq. (1). In the simplest case let $J = J(V(T))$ be some scalar function of the terminal solution value $V(T)$ (the approach extends straightforwardly to other functionals). Setting $\frac{\mathrm{d}J}{\mathrm{d}V} = a_V(t)$ and $\frac{\mathrm{d}J}{\mathrm{d}\theta} = a_\theta(0)$ where $a_V : [0, T] \to \mathbb{R}^n$ and $a_\theta : [0, T] \to \mathbb{R}^p$ are the solutions to the following adjoint ODE system

$$(a_V)_t = -a_V(t)^T \frac{\partial f}{\partial V}(t, V, \theta), \qquad a_V(T) = \frac{\mathrm{d}J}{\mathrm{d}V(T)}, \tag{18a}$$

$$(a_\theta)_t = -a_V(t)^T \frac{\partial f}{\partial \theta}(t, V, \theta), \qquad a_\theta(T) = 0. \tag{18b}$$

Note that the adjoint system requires the forward solution to have already been computed and that the adjoint system runs backwards in time, i.e. evolving from known states $a_V(T)$ and $a_\theta(T)$ at terminal time $t = T$ to the starting time $t = 0$. Once $a_\theta(0)$ has been computed, the required gradient of the functional $\frac{\mathrm{d}J}{\mathrm{d}\theta} = a_\theta(0)$ can be computed straightforwardly. This continuous adjoint ODE approach can be arbitrarily composed with JAX's programme level automatic differentiation capabilities, mean-





ing that it is possible to add non-ODE based components (smoothers etc.) to the model and use our framework for computing

marginal likelihoods.

## 3    Results and discussion

The purpose of this section is to test the accuracy of REpHMC in calculating the BF by employing it to solve benchmark

problems with complex distributions but well known log marginal likelihoods and thus the BF. We illustrate that the BF can

distinguish between models with an equally good fit by calculating the BF of synthetic discharge data for three different models,

among which is the data generating model. We repeat the experiment using another data generating model. Finally, the BF is

applied to the real-world discharge data.

### 3.1    Gaussian shells example

This section aims to show that the the proposed methodology accurately estimates the marginal likelihood of a synthetic

example. In addition, it illustrates the effectiveness of REpHMC in sampling multimodal distributions. The benchmark example

is the Gaussian shells (Feroz et al., 2009; Allanach and Lester, 2008). This example has two wholly separated Gaussian shells,

making it difficult to sample from. This example has been used to test various techniques for calculating the marginal likelihood

(Thijssen et al., 2016; Henderson and Goggans, 2019). The Gaussian shell likelihood is given as

$$\ell(\theta) = \frac{1}{\sqrt{2\pi w_1^2}} \exp\left[-\frac{(||\theta_1 - c_1|| - r_1)^2}{2w_1^2}\right] + \frac{1}{\sqrt{2\pi w_2^2}} \exp\left[-\frac{(||\theta_2 - c_2|| - r_2)^2}{2w_2^2}\right]. \tag{19}$$

The unknown parameters are $\theta = (\theta_1, \theta_2)$, while the marginalised fixed parameters are $r_1, r_2, w_1, w_2, c_1$ and $c_2$. The first shell

has a radius of $r_1$ and the second shell $r_2$. The first shell is centred at $c_1$ while the second is centred at $c_2$. The variance

(width) of the first shell is $w_1$, and that of shell two is $w_2$. We assign uniform priors to $\theta_1$ and $\theta_2$ in the range -6 to 6 and the

marginalised parameters are set to $w_1 = w_2 = 0.1, r_1 = r_2 = 2, c_1 = 3.5, c_2 = -3.5$. We used 20 temperature schedules, since

this is a difficult sampling problem to obtain fast mixing due to the two regions of probability mass. A plot of the samples for the

parameters using various samplers is shown in Fig. 3. The plot shows that REpHMC can sample across the shells compared to

No-U-Turn sampler (NUTS) and Metropolis-adjusted Langevin algorithm (MALA) which is also a gradient-based algorithm.

The results of the marginal likelihood up to 30 dimensions are shown in Table 2 with agreement with the marginal likelihood

values reported in the literature (Feroz et al., 2009).

### 3.2    Synthetic examples

In this section we generate synthetic discharge data by using the observed precipitation and observed potential evapotranspira-

tion as inputs to our models. The following two examples aim to verify the correct implementation and study the behaviour of

the methodology to calculate the marginal likelihood. In the first experiment, data $y_{obs}$ is generated from the simplest model,

$M_2$. In the second experiment, $M_3$ (three buckets model) is the data generating model. For each experiment, the log-marginal





(a) Posterior samples obtained by NUTS.

(b) Samples obtained by REpHMC.

(c) Samples obtained by MALA.

(d) Target distribution of two dimensional Gaussian shells.

**Figure 3.** Top left is (a) is NUTS, top right (b) is REpHMC, bottom left (c) is MALA and bottom right (d) is the target distribution. Replica Exchange preconditioned Hamiltonian Monte Carlo (REpHMC) samples the entire distribution space in contrast to the No-U-Turn Sampler (NUTS) and Metropolis-adjusted Langevin algorithm (MALA). NUTS is an extension of Hamiltonian Monte Carlo (HMC) sampler that requires no tuning of the number of leapfrog steps $L$.





**Table 2.** Log marginal likelihood (log $p(y)$) of the Gaussian shell example. The true values are shown, and the estimates are based on thermodynamic integration with samples from REpHMC. The results are shown for up to 30 dimensions.

| Dimensions | *Reference log $p(y)$ | Estimated log $p(y)$ |
|---|---|---|
| 2 | -1.75 | $-1.75 \pm 0.003$ |
| 5 | -5.67 | $-5.68 \pm 0.006$ |
| 10 | -14.59 | $-14.60 \pm 0.006$ |
| 20 | -36.09 | $-36.12 \pm 0.014$ |
| 30 | -60.13 | $-60.19 \pm 0.025$ |

* As reported in Feroz et al. (2009)

likelihood $\log p(y|M_i)$ for $i = 2, 3, 4$ and the respective Bayes factors are calculated. The deviance information criterion (DIC) and widely applicable information criterion (WAIC) are also calculated for experiments in Section 3.2.1, Section 3.2.2 and for real-world discharge data in Section 3.3.

### 3.2.1 Experiment one with data generated from the two-buckets model $M_2$

In the first experiment, synthetic discharge data $y_{\mathrm{obs}}$ is generated from the simplest model, $M_2$ (two buckets model) to see if the BF will select $M_2$. We set up the priors as in Table 3. The synthetic discharge is generated to have similar dynamics as the observed discharge shown in Fig. 4. First, we obtain the daily precipitation and evapotranspiration for the Magela Creek catchment in Australia for 1980. The initial time $t = 0$ corresponds to midnight on January 1, 1980, and the final time $T = 366$ days to midnight on December 31, 1980 (1980 had a leap year). It is assumed that the total precipitation and evapotranspiration on a given day is uniformly distributed over the 24 hours from midnight to midnight. This is an acceptable assumption when modelling the dynamics of a catchment on a multiday time scale.

Our analysis focuses on a three-month period in 1980. This period runs from 01-01-1980 to 31-03-1980, as represented in Fig. 5, when the precipitation frequency is highest, and there are no missing data.

We set up the priors according to the following reasoning:

– The top bucket associated with state $V_1$ typically represents the fast dynamics of the catchment system, such as surface runoff into rivers. The parameters $k_1$ and $k_{12}$ are the recession coefficients of the top bucket. They represent the flow rates from the top bucket. Since the parameters have to be positive, we use lognormal priors, the most commonly used distribution for dynamic models.

– The lower buckets represent processes with progressively slower dynamics, such as groundwater storage. The parameters $k_i$ and $k_{(i-1),(i)}$ are the recession coefficients for the $n^{th}$ bucket with $i = 2, 3, \dots, n$.

– The system starts with a nonzero initial condition that mimics the standard procedure of "bootstrapping" the ODE system for a period $T_B < 0$. For real-world data, the initial conditions are not known and must be identified. The initial condition to be identified is $\hat{V}_i$ where $i = 1, 2, \dots, n$.





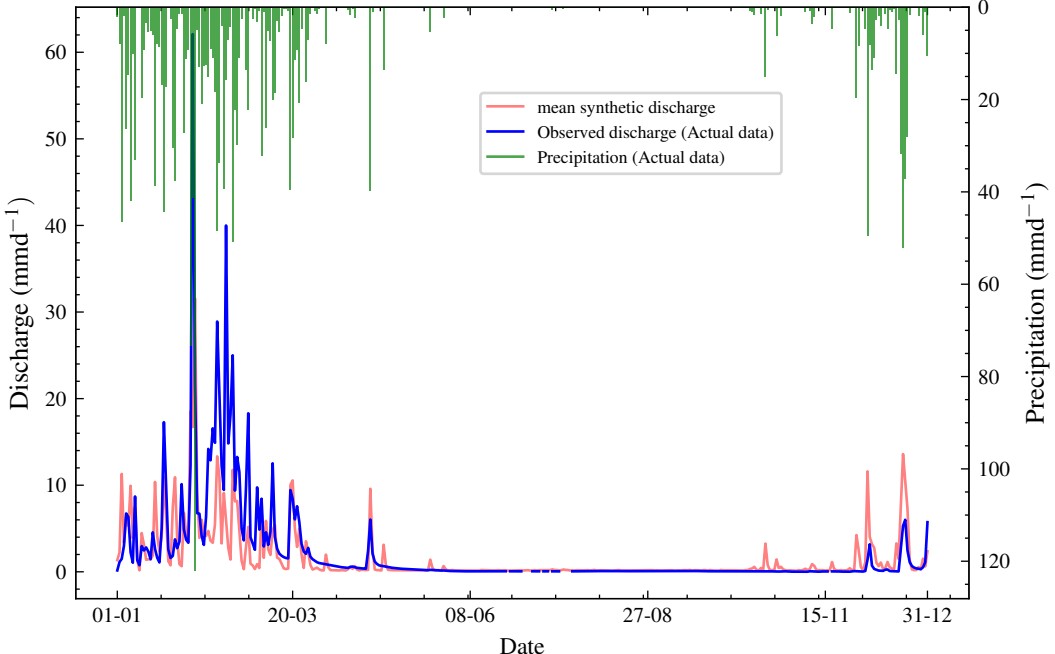

**Figure 4.** Plot of observed discharge, synthetic discharge, and precipitation from 01-01-1980 to 31-12-1980. The observed discharge has missing values, represented by the broken blue line, mostly in the seventh month. Synthetic discharge data generated via the joint posterior (before calibration) shows similar overall trends to the observed discharge.

The meaning of the parameters and the priors are shown in Table 3. We follow a Bayesian workflow and do a prior predictive check. This helps to verify if the priors are reasonable. For the prior predictive check, 50 samples were drawn from the prior and then evaluated in the likelihood. This gave 50 different data sets for the synthetic discharge. The mean synthetic discharge

is then obtained, and the 95 % pointwise credible intervals are obtained and shown in Fig. 6. The marginal likelihoods for $M_2, M_3$ and $M_4$ were calculated and the corresponding Bayes factors were calculated. For each model, fifteen different runs of the marginal likelihood were calculated using REpHMC + TI. This enabled us to get the estimate's standard deviation, which is different from the Monte Carlo standard error.

     We perform REpHMC with 10 replicas where the likelihood of a replica is raised to an inverse temperature value according

to the schedule in Eq. (14). Each replica was run until IAT $< N/50$, where $N$ is the number of posterior samples. The IAT is the number of samples required to obtain an independent sample and a smaller value is preferable. We found that 4000 posterior samples per replica were enough to rule out non-stationarity. We also did a full run with 20000 posterior samples per chain, and we saw no significant change in the results. The p-value for Geweke diagnostics was not significant at 5 % for all parameters and models (p-value $> 0.90$), indicating there is a high probability that the parameters have converged. The

IAT and Geweke diagnostics were performed using the Python package, pymcmcstat (Miles, 2019). The posterior parameter estimates and 95 % credible interval (CI) are in table Table 4. For $M_2$, the true model, the posterior parameters are very close to the true values and are within the 95 % CI. Moreover, the parameters $k_1, \hat{V}_1, \hat{V}_2, V_{\max}$ and $\sigma^2$ are very close to the true



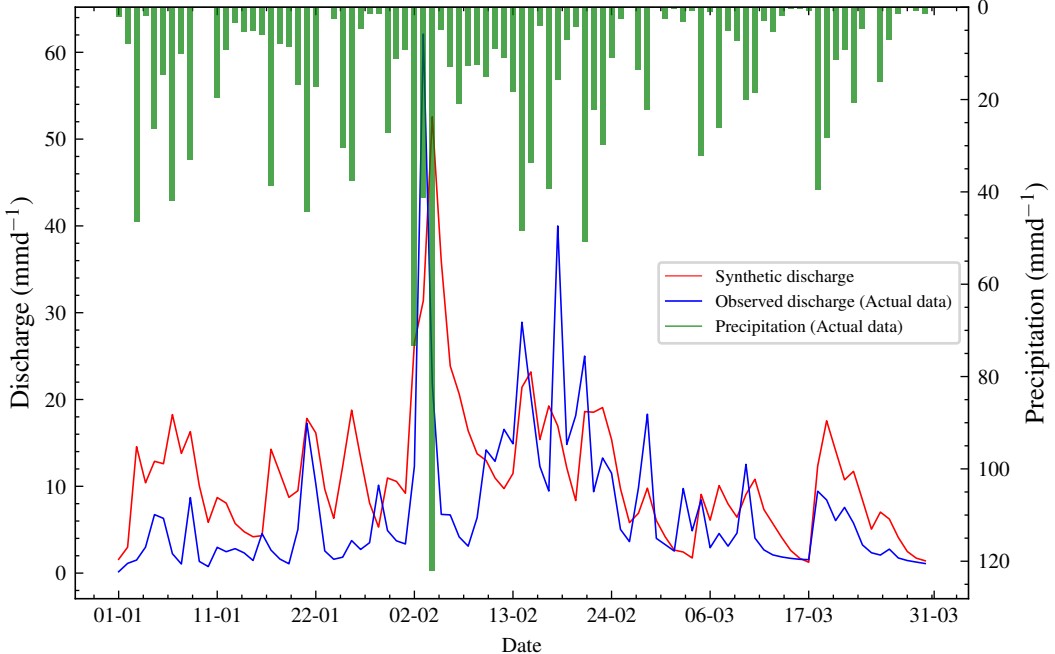

**Figure 5.** Plot of observed discharge, synthetic discharge, and precipitation from 01-01-1980 to 31-03-1980. This period has no missing values and has the highest precipitation frequency and discharge of the year 1980. The synthetic discharge has a similar trend to the observed discharge. The synthetic discharge here is generated using a different set of parameters compared to that in Fig 4.

**Table 3.** Description of the parameters and priors. Note that here we have used units more common in the hydrological literature. LN is the lognormal distribution and IG is the inverse Gamma distribution. The IG was chosen because it is easier to sample than other distributions for the prior noise parameter, which must be positive.

| Parameter | Unit | Description | Prior |
|:---------:|:----:|:-----------:|:-----:|
| $k_1$ | $\text{d}^{-1}$ | Outflow recession coefficient for bucket 1 | $\text{LN}(1.0, 0.25)$ |
| $k_2$ | $\text{d}^{-1}$ | Outflow recession coefficient for bucket 2 | $\text{LN}(0.6, 0.25)$ |
| $k_3$ | $\text{d}^{-1}$ | Outflow recession coefficient for bucket 3 | $\text{LN}(0.3, 0.25)$ |
| $k_4$ | $\text{d}^{-1}$ | Outflow recession coefficient for bucket 4 | $\text{LN}(0.1, 0.25)$ |
| $k_{12}$ | $\text{d}^{-1}$ | Interbucket recession coefficient 1 to 2 | $\text{LN}(0.8, 0.25)$ |
| $k_{23}$ | $\text{d}^{-1}$ | Interbucket recession coefficient 2 to 3 | $\text{LN}(0.4, 0.25)$ |
| $k_{34}$ | $\text{d}^{-1}$ | Interbucket recession coefficient 3 to 4 | $\text{LN}(0.1, 0.25)$ |
| $\hat{V}_1$ | mm | Initial condition on $V_1$ | $\text{LN}(0.0, 1.0)$ |
| $\hat{V}_2$ | mm | Initial condition on $V_2$ | $\text{LN}(0.0, 1.0)$ |
| $\hat{V}_3$ | mm | Initial condition on $V_3$ | $\text{LN}(0.0, 1.0)$ |
| $\hat{V}_4$ | mm | Initial condition on $V_4$ | $\text{LN}(0.0, 1.0)$ |
| $V_{\max}$ | mm | Maximum amount of water the soil can store | $\text{LN}(1.0, 0.25)$ |
| $\sigma^2$ | $\text{mm}^2\text{d}^{-2}$ | Variance of the Gaussian noise model | $\text{IG}(5.0, 0.1)$ |





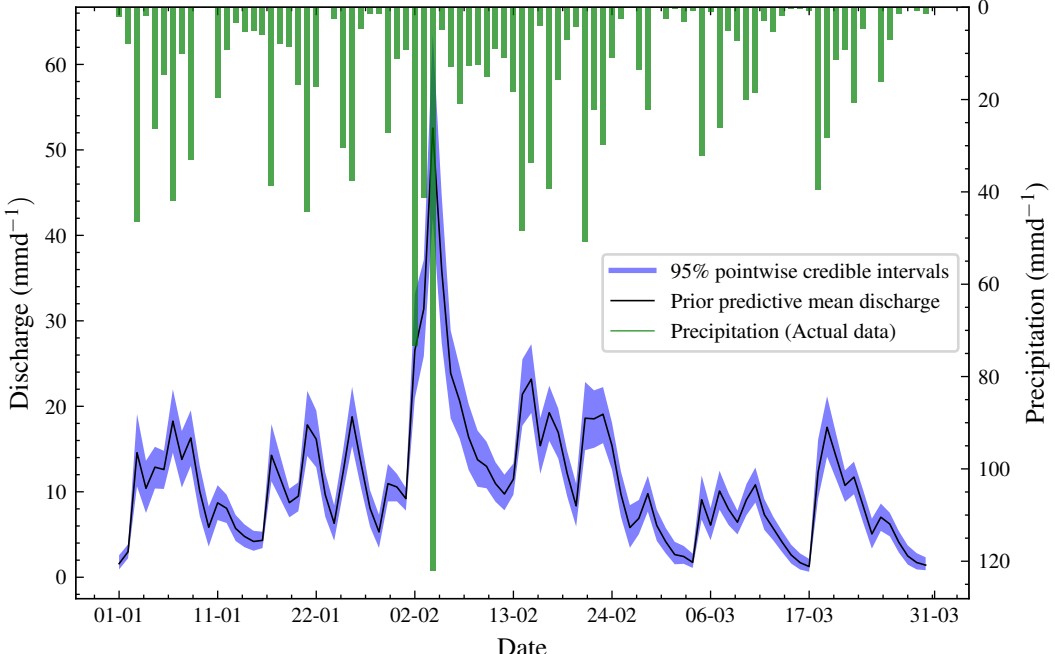

**Figure 6.** Prior predictive check for synthetic data. Fifty prior samples were used to obtain fifty different discharge data sets. The synthetic mean discharge and the 95 % pointwise confidence intervals are shown.

values. However, the error term $\sigma^2$ is the same for all three models, as all models fit the data well. Therefore, a model selection criterion is needed to discriminate between models. Fifteen marginal likelihoods are calculated in parallel for each model.

The mean log marginal likelihood is presented in Table 4. We can calculate the log BF of any model compared to another by taking the difference in their log marginal likelihoods. Based on the interpretation of BF in Table 1, there is decisive evidence in favour of the data generating model $M_2$. The distributions of the log marginal likelihood for each model are shown in box plots (Fig. 7a). In addition, the DIC and WAIC are shown along with those of the marginal likelihood and they also select the data generating model. Model $M_2$ has the largest median log marginal likelihood, while model $M_4$ has the lowest. The

prior and posterior distributions for model $M_2$ are in Fig. 8. The prior distribution is in blue, while the posterior is in red. The prior range is wide compared to the posterior such that the posterior contours are too small. The posterior marginal densities are also more contracted compared to the prior densities, as seen on the diagonal of the plots. The prior contours show no significant correlation between the parameters. The posterior distributions for this model are shown in Fig. 9. The marginal posterior distributions are on the diagonal. The red dots represent the true parameters. There is also a high correlation between

pairs $(k_1, k_2), (k_1, V_{\max}), (k_{1,2}, k_2), (k_{12}, V_{\max}), (k_2, V_{\max})$ and $(\hat{V}_1, \hat{V}_2)$.

We also performed graphical posterior predictive checks. Discharge data was generated from the posterior predictive distribution of each model and plotted. There is no noticeable visual difference in discharge (Fig. 10) for all the models since the





**Table 4.** True value, posterior mean with 95 % credible intervals of the parameters, and log marginal likelihood of the models for experiment one. Model $M_2$ has the highest log marginal likelihood and is the true model. The DIC and WAIC are also shown.

| parameter | True value | $M_2$ (95 % CI) | $M_3$ (95 % CI) | $M_4$ (95 % CI) |
|---|---|---|---|---|
| $k_1$ | 1.454 | 1.454 (1.445, 1.462) | 1.438 (1.434, 1.457) | 1.089 (1.081, 1.095) |
| $k_2$ | 0.248 | 0.248 (0.248, 0.248) | 0.241 (0.241, 0.250) | 0.160 (0.129, 0.174) |
| $k_3$ | 0.000 | - | 0.248 (0.247, 0.248) | 0.241 (0.196, 0.265) |
| $k_4$ | 0.000 | - | - | 0.208 (0.207, 0.208) |
| $k_{1,2}$ | 3.232 | 3.234 (3.205, 3.263) | 3.157 (3.145, 3.256) | 1.628 (1.552, 1.670) |
| $k_{2,3}$ | 0.000 | - | 1.619 (0.993, 1.683) | 1.102 (0.921, 1.400) |
| $k_{3,4}$ | 0.000 | - | - | 1.861 (1.105, 2.749) |
| $\hat{V}_1$ | 1.081 | 1.067 (1.039, 1.095) | 1.067 (1.038, 1.071) | 1.246 (1.181, 1.282) |
| $\hat{V}_2$ | 0.813 | 0.894 (0.787, 0.990) | 0.490 (0.483, 0.593) | 0.599 (0.474, 0.761) |
| $\hat{V}_3$ | 0.000 | - | 0.520 (0.453, 0.525) | 0.731 (0.459, 0.827) |
| $\hat{V}_4$ | 0.000 | - | - | 0.576 (0.433, 0.954) |
| $V_{\max}$ | 2.520 | 2.520 (2.502, 2.542) | 2.573 (2.507, 2.581) | 3.106 (2.999, 3.149) |
| $\sigma^2$ | 0.014 | 0.014 (0.011, 0.016) | 0.015 (0.015, 0.016) | 0.023 (0.022, 0.027) |
| $\log p(y\|M)$ | - | 217.968 | 203.383 | 154.768 |
| DIC | - | -521.235 | -448.980 | -449.000 |
| WAIC | - | -514.354 | -501.686 | -445.233 |

-: The parameter is not included in the model.
$\sigma^2$: error term.
$\log p(y|M)$: log marginal likelihood.

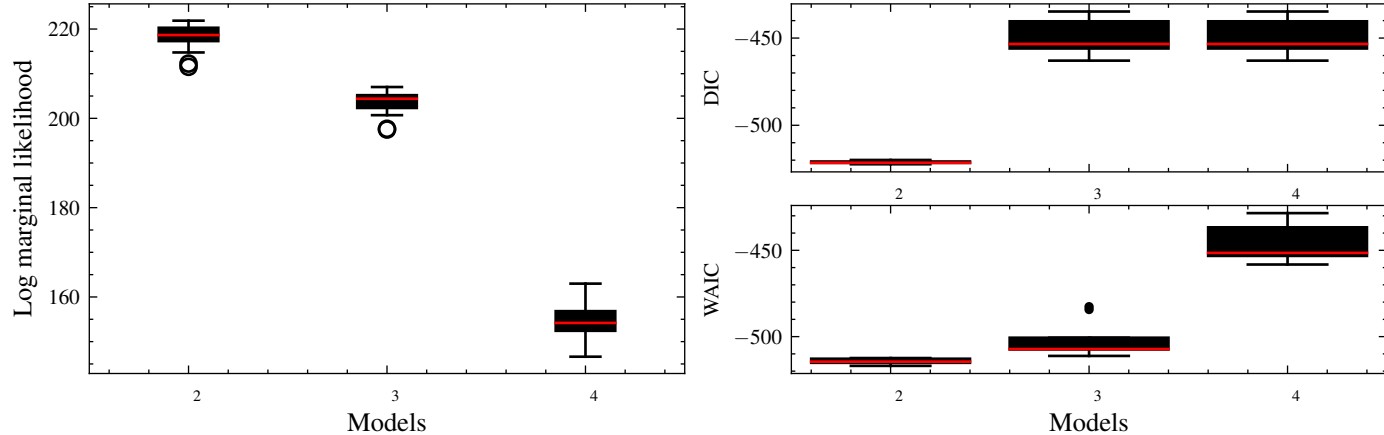

(a) Distribution of the log marginal likelihood for 15 different runs. The boxplot of the data degenerating model, $M_2$, is the highest while $M_4$ is the lowest. Hence, $M_2$ has the highest marginal likelihood. $M_3$ has the shortest interquartile range and, therefore, variability.

(b) DIC and WAIC for 15 model runs.

**Figure 7.** For the log marginal likelihood, higher values are preferred, while for the deviance information criterion (DIC) and widely applicable information criterion (WAIC), smaller values are preferred. All techniques select the data-generating model.





**Figure 8.** Posterior distributions for model $M_2$. It is difficult to see the correlations due to the high difference in variance between the prior and posterior distributions. The red represents the posterior distributions and the blue the prior distributions. The posterior distributions have contracted compared to the priors.

posterior error estimate is too small for all models. We also calculated PPP for the selected model using autocorrelation as a discrepancy measure. Hence, Eq. (8) becomes


$$\text{ppp}(y_{\text{obs}}) = \frac{1}{n}\sum_{i=1}^{n} I\big[(\rho_i^{\text{rep}},\theta_i) \geq (\rho_{\text{obs}},\theta_i)\big] \tag{20}$$





**Figure 9.** Posterior distributions for model $M_2$. There is a high correlation between $k_1$ and $V_{max}$, $k_{1,2}$ and $k_2$, $k_{1,2}$ and $V_{max}$. The marginal posterior distributions are on the diagonal. The black dots represent the true parameters used in the data generating process.

Posterior predictive plots might not tell us if the chosen model fits the data well, especially for dense data sets. Therefore, formal posterior predictive tests based on the discrepancy measure are needed. Like most statistical tests, the results will depend on the type of discrepancy measure or the test statistics. Carefully choosing such discrepancy measures is crucial. For example,





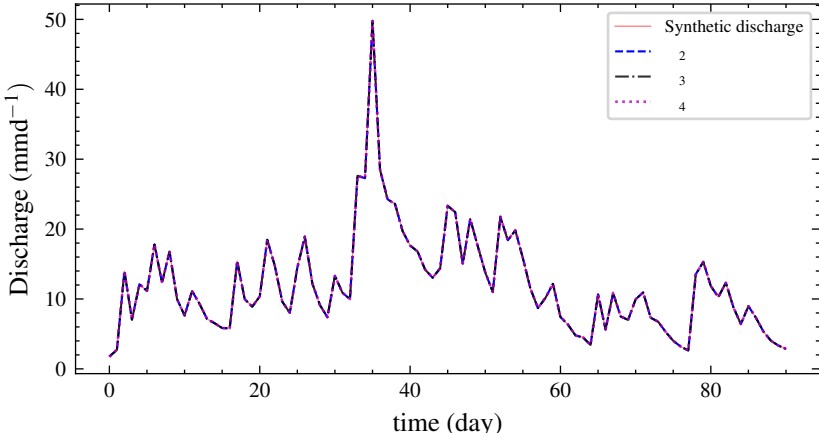

**Figure 10.** Hydrograph generated from the posterior predictive distribution of each model. It is difficult to choose one model by inspection as they all fit the data equally. Hence, BF penalises models with more parameters.

we may test whether the model can predict peak discharge values, which would require a different discrepancy measure than if

the aim of our analysis was to predict the mean values. Hence, we suggest using formal posterior predictive tests and graphical posterior predictive checks as in this study. The PPP is 0.51, which means that the model has good predictive performance. This is expected for synthetic data. Values further from 0.50 indicate a model mismatch with the data. Values closer to zero indicate that the model predictions are lower than the observed data. In contrast, values closer to one point that predictions are higher than observed data. A plot of the autocorrelations of predicted versus synthetic observed data is shown in Fig. 11. The

proportion of values above the $45^\circ$ line is the PPP. We also calculated PCPPP for the selected model and got a value of 0.64 > 0.05, which implies the model is can generate the data. The PCPPP is calibrated based on the prior predictive distribution and is uniformly distributed. Thus, it has the same interpretation as a classical p-value.



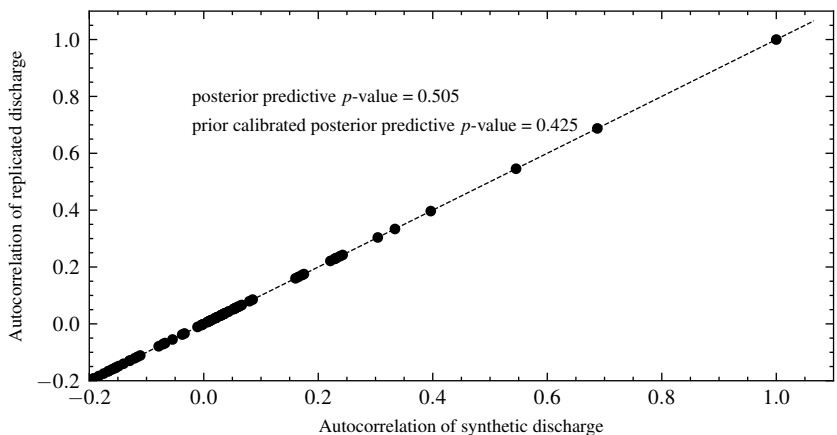

**Figure 11.** Autocorrelation of the replicated versus observed synthetic discharge data. The posterior predictive p-value is the proportion of observations above the 45° line. The autocorrelation of the first point is 1, which isolates it from the other observations.





### 3.2.2 Experiment two with data generated from the three-buckets model $M_3$

For the second experiment, the data model is $M_3$. The model $M_3$ has three more parameters than $M_2$ and three fewer parameters
than the model $M_4$. The priors for model $M_2$ and $M_3$ are shown in Table 3. The data in this experiment was also generated
to follow the same trend as the observed data. All models were fitted to the data, and inference is based on 20,000 posterior
samples with a burn-in of 5,000. As explained above, convergence was checked using IAT and Geweke diagnostics. The
posterior estimates are in Table 5. Although the error term is small for all models, $M_2$ has a higher value than the other two
models, suggesting that it may not have the right complexity. Fifteen marginal likelihoods were also calculated for each model
in parallel. The mean log marginal likelihood is presented in Table 3. The results are also in box plots of Fig. 12a. The box
plots reveal that $M_3$ has the highest median log marginal likelihood, and $M_2$ the lowest. There is decisive evidence in favour
of model $M_3$, the expected result. The DIC and the WAIC do not show a preference for the true model ($M_3$) over $M_4$. A
difference of at least 5 is usually required for one to prefer any model and a difference of at least 10 for strong preference in
any model. The differences in this experiment are less than 5. The box plots in Fig. 12a show that the DIC and WAIC are very
close for $M_3$ and $M_4$. Like in the previous experiment, a hydrograph from the posterior predictive distribution is shown in
Fig. 13. From the hydrograph, we cannot determine the best model through visual inspection since all the models fit the data
equally well. Therefore, we require a formal model selection technique like the BF.

**Table 5.** True value, posterior mean with 95 % credible intervals of parameters and log marginal likelihood of models for experiment two.
$M_3$ the true model has the highest log marginal likelihood. The DIC and WAIC are also included.

| parameter | true value | $M_2$ (95 % CI) | $M_3$ (95 % CI) | $M_4$ (95 % CI) |
|:---:|:---:|:---:|:---:|:---:|
| $k_1$ | 1.091 | 1.109 (1.104, 1.113) | 1.090 (1.084, 1.097) | 1.089 (1.081, 1.095) |
| $k_2$ | 0.188 | 0.207 (0.206, 0.207) | 0.172 (0.160, 0.190) | 0.160 (0.129, 0.174) |
| $k_3$ | 0.208 | - | 0.208 (0.207, 0.208) | 0.241 (0.196, 0.265) |
| $k_4$ | 0.000 | - | - | 0.208 (0.207, 0.208) |
| $k_{1,2}$ | 1.675 | 1.772 (1.759, 1.786) | 1.648(1.613, 1.693) | 1.628 (1.552, 1.670) |
| $k_{2,3}$ | 1.050 | - | 1.520 (1.070, 1.781) | 1.102 (0.921, 1.400) |
| $k_{34}$ | 0.000 | - | - | 1.861 (1.105, 2.749) |
| $\hat{V}_1$ | 1.317 | 1.263 (1.224, 1.325) | 1.302 (1.242, 1.346) | 1.246 (1.181, 1.282) |
| $\hat{V}_2$ | 0.936 | 1.758 (1.622, 1.914) | 0.977 (0.733, 1.167) | 0.599 (0.474, 0.761) |
| $\hat{V}_3$ | 0.910 | - | 0.856 (0.696, 1.103) | 0.731 (0.459, 0.827) |
| $\hat{V}_4$ | 0.000 | - | - | 0.576 (0.433, 0.954) |
| $V_{\max}$ | 3.048 | 2.929 (2.910, 2.948) | 3.081 (3.026, 3.127) | 3.106 (2.999, 3.149) |
| $\sigma^2$ | 0.024 | 0.027 (0.024, 0.030) | 0.023 (0.020, 0.027) | 0.023 (0.022, 0.027) |
| $\log p(y|M)$ | - | 161.586 | 173.845 | 148.060 |
| DIC | - | -401.612 | -427.913 | -426.127 |
| WAIC | - | -394.247 | -420.380 | -417.174 |

-: The parameter is not included in the model.
$\sigma^2$: error term.
$\log p(y|M)$: log marginal likelihood.



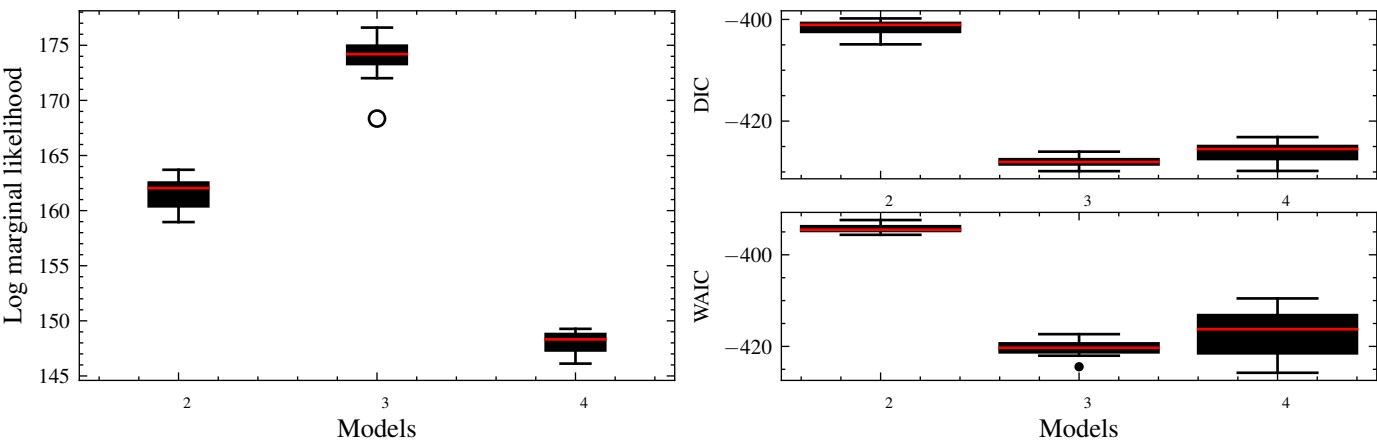

(a) Distribution of the log marginal likelihood for 15 model runs with different initial parameter values. $M_3$, the data generating model has the highest median, while $M_4$ has the lowest. $M_4$ has the highest number of parameters, while $M_2$ has the least.

(b) DIC and WAIC for 15 model runs.

**Figure 12.** For the log marginal likelihood, higher values are preferred, while for the DIC and WAIC, smaller values are preferred. The log marginal likelihood selects the data-generating model, while DIC and WAIC do not have any preference for model $M_3$ and $M_4$.

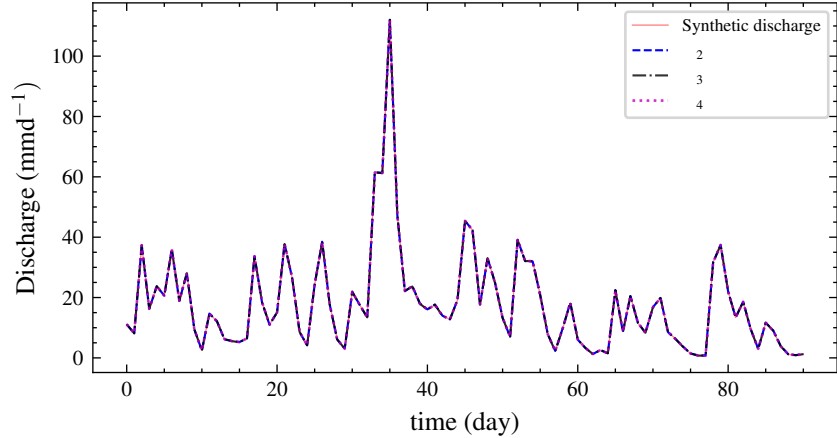

**Figure 13.** Hydrograph from the posterior predictive distributions show a good fit for all models. So it is not possible to select one by inspection. However, the BF could select the data generating model.



### 3.3 Real data experiment

This section uses real-world discharge data for Magela Creek in Australia. For each model, 10 chains of the REpHMC were run
as in the previous examples. We obtained 4000 posterior samples per chain, discarding the first 1000 as burn-in. The trace plots
showed no indication of non-stationarity of the Markov chain, and both Geweke diagnostics and IAT supported convergence.
The Z-statistic, p-value, and IAT are shown in Table 7. All p-values are greater than 0.05, indicating no significant difference
in the means of earlier and later posterior samples and no evidence against convergence. The null hypothesis states that the
mean of the earlier and later posterior samples are equal. Furthermore, the IAT is less than $N/50$ for all parameters, indicating
well-mixed and stationary chains, where $N$ represents the number of posterior samples. Smaller values of IAT indicate that
fewer samples are needed to obtain an independent sample in the Markov chain.

Since we do not use an objective Bayesian approach, we used two sets of priors, where the second set is a sensitivity
analysis. The first set of priors has higher variances for some parameters and is less informative than the second set (Table 6). It
is common practice to try different priors and to check if the parameter estimates change with different priors. This is known as
prior-sensitivity analysis. The models converge faster with the second set of priors. The first set of priors (Table 3) is the same
as in the previous sections. For the second set of priors, we used lognormal priors with lower variances for some parameters
compared to the first set of priors. The mean values used for the priors are also different from those of the first set of priors.
The prior to the error term remains unchanged.

**Table 6.** Second set of priors. LN is the lognormal distribution and IG is the inverse Gamma distribution

| Parameter | Prior distribution |
| :---: | :---: |
| $k_1$ | LN(0.8, 0.25) |
| $k_2, k_3, k_4$ | LN(0.2, 0.25) |
| $k_{12}, k_{23}, k_{34}$ | LN(0.6, 0.25) |
| $\hat{V}_1, \hat{V}_2, \hat{V}_3, \hat{V}_4, V_{\max}$ | LN(0.0, 0.25) |
| $\sigma^2$ | IG(5.0, 0.1) |

We checked the precision of our chosen model by comparing predicted and observed discharges using a posterior predictive
check based on a second set of priors. The observed and predicted discharge plots are shown in Fig. 16. The plots of the
predicted and observed autocorrelations with PPP are in Fig. 17. The PPP is 0.395 which is not too close to 0.5 and the PCPPP
is 0.511. Hence, one can conclude that the model fits the data based on autocorrelation. Instead of autocorrelation, another
metric could be used for the posterior predictive check depending on the objective of the model. The Nash Sutcliffe efficiency
(NSE) for the chosen model is 0.397 and the Kling Gupta efficiency (KGE) is 0.524. This means that the model performs
better than using the mean observed discharge. Knoben et al. (2019) found that the KGE is < -0.41 when the model performs
poorer than the mean observed discharge. The marginal posterior distributions for the model $M_4$ are shown in Fig. 14. We have
also presented the posterior distributions of the parameters in model $M_3$ in Fig. 15. There is no noticeable correlation between




parameters when real-world discharge data is used. However, $V_{\max}$ plays a major role in the dynamics of the model. A more realistic prior for $V_{\max}$ based on the soil physics of Megala Creek Australia will reduce the model error.

The results of the second set of priors are in Table 8. The selected model did not change when we used diffuse priors. The error in the second set of models is lower than in the first set. The model $M_3$ is always preferred while $M_4$ is always the least supported by the data. The error term, its precision, effective sample size (ESS), and the number of parameters influence the marginal likelihood. We also applied two fully Bayesian information criteria, DIC and WAIC. Unlike BF, all these information criteria preferred $M_4$. However, the difference in DIC or WAIC between $M_3$ and $M_4$ is less than 10.

The RWM, NUTS and MALA were also applied to all the three models with real world data. Even the other gradient-based algorithms NUTS and MALA could not sample the parameter space. Attempts to improve algorithms by trying various values for the initial step size in the case of NUTS and the step size for MALA did not make any difference. This further confirms the fact that combining replica exchange with an algorithm improves mixing and convergence.

**Table 7.** Convergence diagnostics for real-world data. Z-statistic, p-value and IAT. The null hypothesis is that the mean of earlier posterior samples is the same as that of later posterior samples in a Markov chain. All p-values are above 0.05, indicating no significant difference in the mean of earlier and later posterior samples and no evidence against convergence. The IAT is the number of samples required to obtain an independent sample in the Markov chain and smaller values are preferred.

| | Model | | | | | |
| | $M_2$ | | $M_3$ | | $M_4$ | |
| parameter | Z-statistic (p-value) | IAT | Z-statistic (p-value) | IAT | Z-statistic (p-value) | IAT |
|---|---|---|---|---|---|---|
| $k_1$ | 0.029 (0.977) | 8.489 | -0.169 (0.866) | 5.561 | -0.001 (0.999) | 4.811 |
| $k_2$ | 0.631 (0.528) | 3.254 | 0.520 (0.603) | 15.302 | 0.221 (0.825) | 16.99 |
| $k_3$ | - | - | -0.432 (0.666) | 14.723 | 0.137 (0.891) | 9.892 |
| $k_4$ | - | - | - | - | -0.371 (0.710) | 8.542 |
| $k_{1,2}$ | 0.136 (0.892) | 21.421 | 0.423 (0.672) | 22.547 | 0.358 (0.720) | 12.855 |
| $k_{2,3}$ | - | - | 0.399 (0.690) | 20.578 | -0.253 (0.800) | 21.233 |
| $k_{3,4}$ | - | - | - | - | 0.291 (0.771) | 9.495 |
| $\hat{V}_1$ | -0.801 (0.423) | 29.976 | 0.084 (0.933) | 7.650 | 0.037 (0.970) | 11.571 |
| $\hat{V}_2$ | -0.809 (0.419) | 40.986 | -0.015 (0.988) | 8.317 | 0.045 (0.964) | 20.099, |
| $\hat{V}_3$ | - | - | -0.226 (0.821) | 15.710 | 0.264 (0.792) | 8.548 |
| $\hat{V}_4$ | - | - | - | - | -0.402 (0.688) | 12.131 |
| $V_{\max}$ | -0.146 (0.884) | 15.897 | -0.184 (0.854) | 5.786 | 0.032 (0.975) | 3.953 |
| $\sigma^2$ | < -0.0001(1.000) | 9.092 | 0.018 (0.985) | 1.761 | 0.001 (1. 000) | 2.167 |

### 3.3.1 Hydrograph of model $M_3$

Based on the hydrograph Fig. 16, most of the model predictions are very close to the observed discharge and within 50 % pointwise credible intervals. However, two peaks are not captured in the model. The first peak discharge period was from 04-02-1980 to 05-02 1980. The observed precipitation during this period is 41.4 mmd$^{-1}$ to 122 mmd$^{-1}$ on 04-02-1980 and





05-02-1980 respectively. The observed discharge on these days is 62.09 mmd$^{-1}$ and 21.82 mmd$^{-1}$ respectively. It is illogical

that the discharge is reduced with similar weather conditions. The second peak event occurred from 19-03-1980 to 20-03-1980.

The precipitation on 19-03-1980 was 39.5 mmd$^{-1}$, and on 20-03-1980, it was 28.3 mmd$^{-1}$ with potential evapotranspiration

similar to other days. This observed discharge is irrational because the discharge from 27-02-1980 (0 mmd$^{-1}$ of precipitation)

to 29-02-1980 (22 mmd$^{-1}$ of precipitation) is 18.3 mmd$^{-1}$. An alternative explanation for the mismatch in peak discharge

could be that the field capacity of the soil changed during these periods and is not captured in our models.

**Table 8.** Posterior summary statistics and log marginal likelihood for models with the second set of priors. Surprisingly, Model $M_3$ is still preferred even though with a slightly higher error term and wider credible intervals. The answer in preference of $M_3$ might lie in the effective sample size (ESS). $M_3$ has a higher effective sample size for the error term than the other two models. The error term in $M_3$ has an ESS of 2647.0, $M_2$ of 546.0, and $M_4$ of 2257.0. Thus, $M_3$ has more information than $M_2$ with fewer parameters and $M_4$ with more parameters. The DIC and WAIC select $M_4$, and the difference between $M_4$ and $M_3$ is smaller than 10 which is usually the threshold for strong preference for one model.

|  | $M_2$ (95 % CI) | $M_3$ (95 % CI) | $M_4$ (95 % CI) |
|---|---|---|---|
| $k_1$ | 1.093 (0.802, 1.385) | 0.926 (0.689, 1.222) | 0.915 (0.601, 1.228) |
| $k_2$ | 0.229 (0.117, 0.337) | 0.253 (0.148, 0.375) | 0.243 (0.133, 0.368) |
| $k_3$ | - | 0.215 (0.120, 0.319) | 0.242 (0.131, 0.358) |
| $k_4$ | - | - | 0.218 (0.124, 0.324) |
| $k_{1,2}$ | 1.052 (0.639, 1.494) | 1.103 (0.744, 1.480) | 1.326 (0.808, 1.889) |
| $k_{2,3}$ | - | 1.841 (0.978, 2.728) | 2.818 (1.791, 4.171) |
| $k_{3,4}$ | - | - | 1.363 (0.785, 1.948) |
| $\hat{V}_1$ | 1.032 (0.575, 1.559) | 1.016 (0.527, 1.496) | 1.017 (0.551, 1.492) |
| $\hat{V}_2$ | 1.146 (0.560, 1.776) | 1.053 (0.534, 1.516) | 1.038 (0.619, 1.609) |
| $\hat{V}_3$ | - | 1.112 (0.663, 1.714) | 1.021 (0.524, 1.521) |
| $\hat{V}_4$ | - | - | 1.084 (0.607, 1.611) |
| $V_{\max}$ | 2.862 (1.820, 4.070) | 2.702 (1.758, 3.882) | 1.449 (0.948 ,2.043) |
| $\sigma^2$ | 6.060 (5.389, 6.763) | 5.916 (5.279, 6.585) | 5.818 (5.143, 6.444) |
| $\log p(y\|M)$ | -3416.325 | -541.647 | -3766.742 |
| DIC | 981.150 | 974.878 | 969.586 |
| WAIC | 990.836 | 982.127 | 975.032 |

-: The parameter is not included in the model.
$\sigma^2$: error term.
$\log p(y|M)$: log marginal likelihood.

## 3.4 Convergence

### 3.4.1 Model convergence time

All models converge by 3000 samples, even for real-world data. The models were run on high-performance computers. The

model runtime of Gaussian shell examples ranges from 6 seconds for 2 dimensions to 24 seconds for 30 dimensions. Syn-

thetic examples converge in 2 to 4 hours, depending on the parameter's dimension. On the contrary, with real data, the models

converge in 6 to 9 hours, depending on the parameter space. Models can converge faster with proper tuning of the number of



leapfrog steps. With large datasets, REpHMC can be combined with subsampling without replacement to accelerate convergence. The REpHMC converges in minutes if we are interested only in parameter estimation. The convergence is even faster for statistical models.

### 3.4.2 Convergence of marginal likelihood

The convergence of the thermodynamic integration estimate of the marginal likelihood depends on the number of inverse
temperatures. The marginal likelihood was stabilised by eight temperature values when calibrating with the real data, but the present results are based on 10 values as recommended by Calderhead and Girolami (2009).





**Figure 14.** Posterior distributions of the 13 parameters for model $M_4$ using the second set of priors. There is no obvious correlation between the parameters. The marginal posterior distributions are on the diagonal.



**Figure 15.** Posterior distributions of the 10 parameters of model $M_3$ based the second set of priors. There is no pronounced correlation between the parameters. The marginal posterior distributions are on the diagonal.



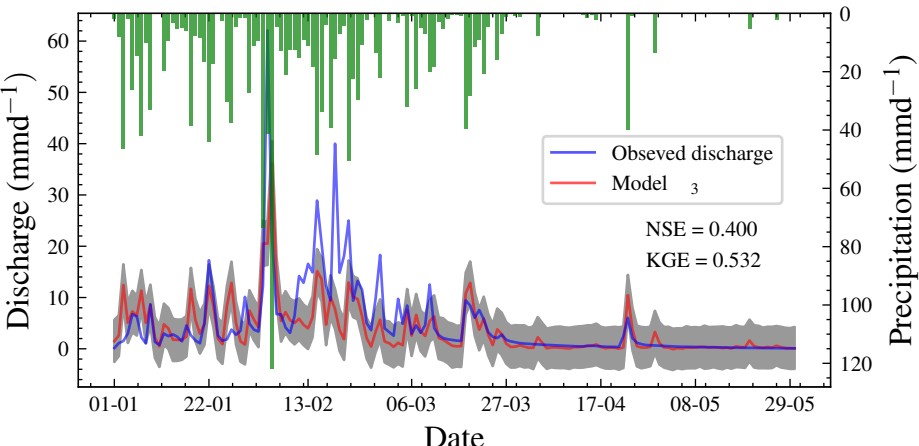

**Figure 16.** Hydrograph based on model $M_3$. The results are better than the prior predictive check shown in Fig. 5, where most predictions are further from the observed data.

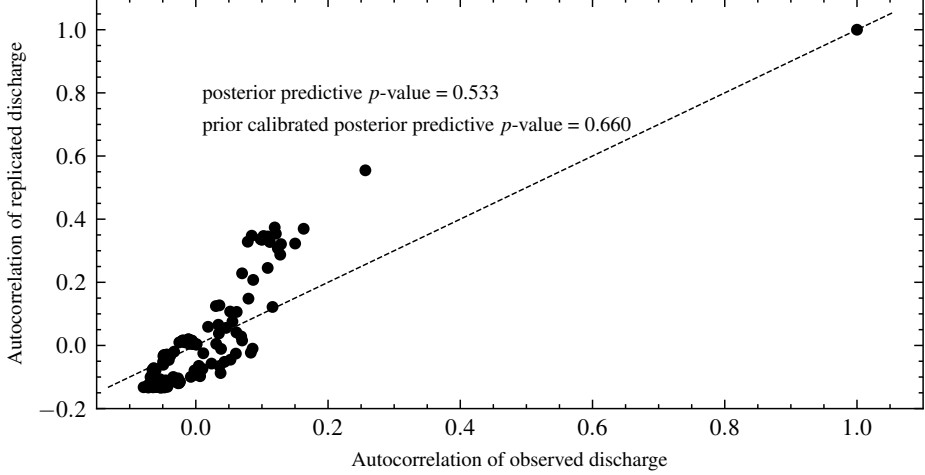

**Figure 17.** Autocorrelation of replicated versus observed data for model $M_3$. The posterior predictive p-value is the proportion of observations above the 45° line.





## 4   Conclusions

We have introduced a methodology for simultaneous Bayesian parameter estimation and model selection. The methodology includes formal model diagnostics, which check for goodness-of-fit and prior data conflict. The method uses a new gradient-based algorithm REpHMC to draw posterior samples, TI for the calculation of marginal likelihood and PCPPP for model diagnostics. The REpHMC and TI were validated on the Gaussian shells example, which is a difficult sampling benchmark problem since it has isolated modes. The REpHMC is effective in sampling the entire parameter space for models with isolated modes. This sets it apart from other gradient-based algorithms such as HMC, NUTS and MALA. Also, we have shown that BF selects the data generating model in two experiments, while DIC and WAIC correctly select the true model in one of two experiments. Also, none of the other mentioned gradient-based algorithms worked when real-world data was used with our developed model. In addition, formal posterior predictive checks have been introduced to determine if a model can accurately predict observed or desired values, such as the minimum or peak discharge. The method was employed to discharge data from Magela Creek in Australia. We also calculated NSE and KGE for the chosen model with real-world data. The framework has been implemented in open-source software TFP which supports most algorithms. The REpHMC can be applied to any hydrological model. Our developed model performed better than using the mean as a predictor for real discharge data. However, the model does not capture peak discharge values. Therefore, some improvements in that regard need to be made.

By combining a gradient-based algorithm HMC and REMC, we get a very powerful algorithm that can sample complex posteriors thanks to the exchange of information between parallel running chains. We have also illustrated that the BF is a reliable Bayesian tool for model selection in contrast to two common Bayesian-based information criteria for model selection.

Future work could combine REMC with NUTS algorithm to automatically tune all parameters in the HMC. Also, introducing subsampling in the case of big data or models with millions of parameters will reduce the inference time. Future work could focus on improving the model goodness-of-fit, as the KGE indicates. Furthermore, one could develop a discrepancy measure for the posterior predictive check to test whether the selected model can capture peak discharge values. On the practical side, this study could be extended to the multi-catchment setting. Also, different types of conceptual hydrological models could be compared using this approach.





**Glossary**

**AIC** Akaike information criteria. 1

**BF** Bayes factor. 1, 2, 3, 4, 8, 17, 19, 22, 26, 28, 29, 31, 37

**BIC** Bayesian information criterion. 1

**CI** credible interval. 20

**DIC** deviance information criterion. 19, 22, 28, 29, 31, 32, 37

**DREAM** differential evolution adaptive Metropolis. 4

**ESS** effective sample size. 31

**HBV** Hydrologiska Byråns Vattenbalansavdelning. 2, 3, 6

**HMC** Hamiltonian Monte Carlo. 2, 4, 12, 13, 14, 15, 16, 37

**IAT** integrated autocorrelation time. 13, 20, 28, 30, 31

**KGE** Kling Gupta efficiency. 30, 37

**MALA** Metropolis-adjusted Langevin algorithm. 17, 18, 31, 37

**MCMC** Markov Chain Monte Carlo. 11, 12

**NSE** Nash Sutcliffe efficiency. 30, 37

**NUTS** No-U-Turn sampler. 17, 18, 31, 37

**ODE** Ordinary differential equation. 3, 15, 16

**ODEs** Ordinary differential equations. 2, 5, 6, 14

**PCPPP** prior calibrated posterior predictive p-value. 3, 9, 26, 30, 37

**pHMC** preconditioned Hamiltonian Monte Carlo. 2, 10, 12, 15

**PMC** population Monte Carlo. 12



**PP**  probabilistic programming. 15

**PPC**  posterior predictive check. 8

**PPL**  probabilistic programming language. 15, 16

**PPP**  posterior predictive p-value. 8, 9, 24, 26, 30

**REHMC**  Replica exchange Hamiltonian Monte Carlo. 9, 12, 13

**REMC**  Replica exchange Monte Carlo. 2, 10, 12, 16, 37

**REpHMC**  Replica exchange preconditioned Hamiltonian Monte Carlo. 2, 3, 10, 12, 13, 17, 18, 19, 20, 30, 33, 37

**RWM**  random walk Metropolis. 4, 12, 14, 31

**TFP**  TensorFlow probability. 2, 15, 16, 37

**TI**  thermodynamic integration. 2, 3, 9, 10, 12, 16, 20, 37

**WAIC**  widely applicable information criterion. 19, 22, 28, 29, 31, 32, 37

*Code and data availability.*  The source code, data, and instructions are available on Zenodo (Mingo and Hale, 2023) and GitHub at https://github.com/DamingoNdiwa/hydrological-model-selection-bayes.

*Author contributions.*  DNM: Conceptualization, Data curation, Formal analysis, Investigation, Methodology, Software, Validation, Visualisation, Writing - original draft, Writing - review & editing. RN: Conceptualization, Formal analysis, Methodology, Writing - review & editing. CL: Conceptualization, Funding acquisition, Formal analysis, Methodology, Supervision, Writing - review & editing. JSH: Conceptualization, Formal analysis, Funding acquisition, Methodology, Project administration, Software, Supervision, Validation, Writing - original draft, Writing - review & editing.

*Competing interests.*  The authors declare no competing interests.

*Acknowledgements.*  We would like to thank Stanislaus Schymanski for his valuable insights which inspired us to undertake this study, for his critical feedback on a draft of this manuscript. This work was funded under the Luxembourg National Research Fund under the PRIDE programme (PRIDE17/12252781) and ATTRACT programme (A16/SR/11254288). The experiments presented in this paper were carried out using the HPC (Varrette et al., 2022) facilities of the University of Luxembourg – see https://hpc.uni.lu.



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
