# Peer review of "Selecting a conceptual hydrological model using Bayes' factors computed with Replica Exchange Hamiltonian Monte Carlo and Thermodynamic Integration"

_EGUsphere, 2023_

## Referee Comment (RC1)

**Review of the manuscript: "Selecting a conceptual hydrological model using Bayes' factors computed with Replica Exchange Hamiltonian Monte Carlo and Thermodynamic Integration" by Damian N. Mingo, Remko Nijzink, Christophe Ley, and Jack S. Hale1**

**General comments**

The paper introduces a complex numerical method for robust model comparison using Bayes Factors. More precisely, the authors propose a pipeline for estimating the marginal likelihood (and consequently, the Bayes Factor) by combining thermodynamic integration with Replica exchange Monte Carlo for power posterior ensemble simulation, and Preconditioned Hamiltonian Monte Carlo (pHMC) for efficient gradient-based sampling. This appears to be one of the first approaches that integrate all these sub-algorithms into one pipeline, in conjunction with an innovative implementation of the methodology within a probabilistic programming framework paired with a differentiable programming language. The paper provides a comprehensive overview of the contributions and related work, as well as an extensive explanation of the methodology and all relevant numerical methods. The authors then discuss the implementation aspects and model before presenting the results. While the results section is well-organized, it left me, as a reader, wanting more. Specifically, the results section does not convincingly demonstrate that the presented methodology effectively addresses the problems mentioned in the introduction. Additionally, certain details in the results section appear to be rushed over and sporadically mentioned without proper references or prior introduction. In conclusion, I would recommend enhancing the results section with more convincing evidence and a clearer exposition of the details before considering the paper for publication.

**Specific comments**

1. In the abstract, the sentence detailing the prior calibrated posterior predictive p-value may be too intricate for readers unfamiliar with the basic concept of p-value and 'posterior predictive p-value.'

2. The introduction's layout, which typically presents background and related works before concluding with the paper's contributions, is more familiar to me. Perhaps swapping sections 1.2 Background and 1.1 Contribution could be considered for a more traditional structure.

3. Bayes Factors (BF) are a crucial component of the paper, yet the formula for calculating them within a multimodal context, beyond just two models, is absent. I expected to find this expression, potentially as an extension of Equation 6

4. The sentence after line 280 stating, 'the samples of the replica with $\beta = 1$ are used to estimate the posterior parameters,' highlights a significant procedure that is not adequately explained.

5. In Algorithm 3, index 'j' iterates from 1 to L, representing the number of leapfrog steps. However, the index 'j' does not appear clearly within the algorithm, leading to potential confusion.

6. The No-U-Turn sampler (NUTS) and Metropolis-adjusted Langevin algorithm (MALA) are suddenly introduced in Section 3.1, without prior mention or any references, and are then used for comparing the results obtained with pHMC. Given their relevance to the results section, introducing these MCMC variants earlier in the Background would enhance the paper's cohesiveness.

7. In Figure 6, the produced prior predictive 95% pointwise confidence interval seems quite narrow, which is unexpected given the variability one would anticipate when sampling from a 13-dimensional (prior) uncertainty space. Additionally, the observed discharge should have been plotted for comparison, to evaluate how well it is bracketed by the prior uncertainty interval.

8. The Deviance Information Criterion (DIC) and Widely Applicable Information Criterion (WAIC) are introduced at the end of Section 3.2 without any explanation or references. In Paragraph 430, the IAT number and Geweke diagnostics are also mentioned without reference. It is unclear if these are assumed to be general knowledge. The placement of these terms is somewhat non-intuitive as they are subsequently used throughout the results section.
9. The report lacks a clear statement regarding the number of forward model runs that were evaluated or needed. Is the correct interpretation that 10*4000 runs were conducted, multiplied by 15 for each model?
10. The results of the synthetic experiments from Sections 3.2.1 and 3.2.2, depicted in Figures 10 and 13, are confusing. The Model 4 with four buckets (M_4) seems to be well calibration with the data originating from a much simpler model.  This raises the question of why the hydrographs of Model 4 aligns so closely with those generated by the 'true' model, which would not be expected.
11. What does the conclusion from Figure 10 mean? "Hence, BF penalizes models with more parameters."  How does one conclude this?
12. In Section 3.3, the authors compare the uncertainty bounds in Figure 16 with a prior-predicted hydrograph from Figure 5. However, the hydrograph in Figure 5 represents only a single random realization from the prior, which seems like an inappropriate comparison. It would be more informative to compare the Monte Carlo mean derived from the prior with the mean hydrograph obtained from the learned posterior. As it stands, Figures 5 and 16 do not seem to be compared on an equitable basis.
13. The results and discussion in the results section have not convincingly demonstrated the ability of the prior calibrated posterior predictive p-value to detect prior data conflicts, a capability that was highlighted in the abstract and introduction.
14. Section 3.4.2, titled "Convergence of marginal likelihood", feels brief and incomplete, as if the discussion in unfinished.
15. The Nash Sutcliffe efficiency (NSE) obtained for the selected model is 0.397, which is low for a model deemed to be calibrated. Typically, NSE values below 0.6 are considered 'low'.
16. The results section does not sufficiently demonstrate the efficacy of the model. The findings presented in Figures 14 and 15 lack in-depth discussion. Although convergence diagnostics for real-world data suggest 'good' outcomes, the presentation falls short of being persuasive. Furthermore, the methodology appears to struggle with definitively identifying the most likely model in real-data scenarios, as indicated by the results in Table 8.
17. It would be beneficial to include a visual comparison, such as hydrographs, of the calibrated models M2 and M4 against the real data to better illustrate their performance.
18. Based on the results of Figure 16, the uncertainty band drawn from the posterior seems wide, even for small streamflow values, which does not give a good hint of adequate parameter estimation / model calibration.

**Technical corrections**

1. Review the formatting and positioning of Equations 1 and 2e.
2. Clarify the paragraph containing lines 105, 110, and 115 to eliminate repetitive information and streamline the content for better readability.
3. Check Equation 2b for a possible typographical error: it should state $(V_2)t =$ and $n=2$, rather than $(V_1)t = \quad n>=2$
4. In Equation 3, there appears to be a typo with $k_{2,1}$, it should likely read $k_{1,2}$.
5. The use of 'p' to denote both the number of uncertain parameters (dimensionality of \Theta) and the number of discrete time steps in sections 2.1 and 2.2.1 is confusing. Consider using distinct notation for these two different concepts.

6. The clarity of the paragraph on lines 210-215 could be enhanced. Simplifying the text and focusing on the key points would help to make the paragraph more comprehensible.
7. Confused by the structure – not sure why section 2.4, titled "Preconditioned Hamiltonian Monte Carlo" is a standalone subchapter. It may be more logical to include it in Section 2.3 "Numerical Methods" along with other algorithm steps.
8. Address the overall layout beginning from page 25 to ensure that the content is well-organized and visually accessible to readers.
9. The caption for Figure 8. "Posterior distributions for model M2" may be missing content. Should it be "Prior and posterior distributions for model M2" to accurately reflect the content of the figure?
10. Revise the description of Figure 10 for precision; it should likely specify that the "mean discharge data was generated from the posterior predictive distribution of each model and plotted," assuming that is the intended meaning.
11. Reformulate the awkward phrasing in line 465 to correct the sentence. It should read "which implies the model can generate the data", removing the extra "is".
12. Correct the reference error in line 475: "The mean log marginal likelihood is presented in Table 3" should be updated to "The mean log marginal likelihood is presented in Table 5" to direct readers to the correct table.

---

## Referee Report (RR1)

**Review of the final manuscript: "Selecting a conceptual hydrological model using Bayes' factors computed with Replica Exchange Hamiltonian Monte Carlo and Thermodynamic Integration" by Damian N. Mingo, Remko Nijzink, Christophe Ley, and Jack S. Hale1**

**General comments**

The manuscript has undergone a significant improvement compared to its previous version. It is now much more coherent and pleasant to read. The extensive explanation of the algorithm and numerical methods remains, but now with a much clearer presentation of the results. In particular, the results are now easier to follow and more cohesive. The authors have addressed all the feedback provided by me and Reviewer #2. Overall, I find the manuscript to be cleaner and more organized, with unnecessary clutter removed or rephrased, making it far easier to understand and engage with.

I recommend accepting the manuscript as it is. However, I am offering a few optional suggestions for potential changes, which the authors may choose to reconsider, though they are not required for acceptance.

**Suggestions:**

1. The Contribution section feels a bit too long, particularly the paragraph starting at line 110, which seems somewhat out of place and could potentially be moved...
2. Consider repositioning Figure 1, I believe it is generally preferred for figures to appear after they are referenced in the text.
3. In Equation 7, the meaning of 'D' is unclear—while it seems to represent a general discrepancy measure, it may be helpful to clarify this by adding a phrase such as, 'To compute the PPP, the chosen discrepancy measure, D, is calculated...'
4. It might be worth considering whether the methods section, starting from 2.3, should be placed in a separate section.
5. I would suggest moving Table 3, which currently appears on page 22, closer to its first reference on page 19, as it is relevant to the data presented in Figure 6.
6. There is some confusion regarding the use of capital 'N,' which is sometimes used to refer to the number of chains or replicas (e.g., Figure 2 and Algorithm 1) and at other times (e.g., page 21, line 445) to the number of posterior samples.
7. I noticed the absence of a concrete BF value, which is central to the paper. While it is mentioned that one can easily compute BF by just subtracting the values of the presented log marginal likelihoods (paragraph 455), I believe it would be beneficial to explicitly include a row in the results tables presenting the BF values, as these are among the most important results of the paper.

8. In section 3.3.1, the hydrographs presented in Figure 16 are analyzed, particularly the two peaks that the model(s) failed to capture, corresponding to the dates 04/05-02-1980 and 19/20-03-1980. However, it seems to me that the first peak not captured by the model's uncertainty band is actually closer to 20-02-1980, which to me, suggests a mismatch between the dates mentioned in the text and the data shown in the figure... though, it might be that I am wrong.
9. I found the last sentence in the conclusion regarding future work somewhat confusing. The sentence suggests combining REMC with NUTS to automatically turn all parameters in the HMC, but to me it seems like this idea was not previously mentioned in the text as a possibility in any form...

---

## Author Response (AR2)

"Selecting a conceptual hydrological model using Bayes' factors computed with Replica Exchange Hamiltonian Monte Carlo and Thermodynamic Integration" by Damian N. Mingo, Remko Nijzink, Christophe Ley, and Jack S. Hale.

We have now implemented the changes requested by the reviewers. We have copied the original discussion, and replied again in green with our changes. Line numbers in the green sections are with respect to the diff-ed version for easy review.

We would like to thank the first anonymous reviewer for their thorough and insightful feedback on our manuscript. We respond (normal text) to the comments (italics) inline below with a plan for producing an improved version for the reviewer and the wider community's consideration. We are also more than happy to discuss specific points again before proceeding with these changes.

**Reviewer 1**

https://doi.org/10.5194/egusphere-2023-2865-RC1
https://doi.org/10.5194/egusphere-2023-2865-AC1

**General comments**

*The paper introduces a complex numerical method for robust model comparison using Bayes Factors. More precisely, the authors propose a pipeline for estimating the marginal likelihood (and consequently, the Bayes Factor) by combining thermodynamic integration with Replica exchange Monte Carlo for power posterior ensemble simulation, and Preconditioned Hamiltonian Monte Carlo (pHMC) for efficient gradient-based sampling. This appears to be one of the first approaches that integrate all these sub-algorithms into one pipeline, in conjunction with an innovative implementation of the methodology within a probabilistic programming framework paired with a differentiable programming language. The paper provides a comprehensive overview of the contributions and related work, as well as an extensive explanation of the methodology and all relevant numerical methods. The authors then discuss the implementation aspects and model before presenting the results. While the results section is well-organized, it left me, as a reader, wanting more. Specifically, the results section does not convincingly demonstrate that the presented methodology effectively addresses the problems mentioned in the introduction. Additionally, certain details in the results section appear to be rushed over and sporadically mentioned without proper references or prior introduction. In conclusion, I would recommend enhancing the results section with more convincing evidence and a clearer exposition of the details before considering the paper for publication.*

**Specific comments**

*1.       In the abstract, the sentence detailing the prior calibrated posterior predictive p-value may be too intricate for readers unfamiliar with the basic concept of p-value and 'posterior predictive p- value.'*

We will remove this point which is too technical for the abstract and introduction - we will also discuss this again in our response to Question 13.

We have removed this sentence from the bottom of the abstract (lines 6-8).

*2.       The introduction's layout, which typically presents background and related works before concluding with the paper's contributions, is more familiar to me. Perhaps swapping sections 1.2 Background and 1.1 Contribution could be considered for a more traditional structure.*

We will switch the subsections around and make some additional changes if needed to maintain the flow of the overall section.

We have switched the order to first show background and then contributions (lines 28-74 & 75-85).

*3.       Bayes Factors (BF) are a crucial component of the paper, yet the formula for calculating them within a multimodal context, beyond just two models, is absent. I expected to find this expression, potentially as an extension of Equation 6*

We will remove the specific case and add the general case for two models indexed with e.g. i and j.

We have added the more general case with i and j subscripts (lines 238 to 241 & Table 1 ).

*4.       The sentence after line 280 stating, 'the samples of the replica with β = 1 are used to estimate the posterior parameters,' highlights a significant procedure that is not adequately explained.*

We agree, this is an important point for readers that was only briefly touched on; although all of the replicas are used to improve chain mixing and in the subsequent marginal likelihood calculation, the parameter estimates are derived only from the statistics of the beta = 1 chain. We will improve this section with comments to this effect.

We have added the following text:

We emphasise that the samples of the replica with $\beta = 1$ are used to estimate the posterior parameters, while the entire ensemble is used as input within TI to calculate the marginal likelihood (lines 339 to 341).

5.      In Algorithm 3, index 'j' iterates from 1 to L, representing the number of leapfrog steps. However, the index 'j' does not appear clearly within the algorithm, leading to potential confusion.

We will correct this issue.

We have removed the explicit index j which isn't used in the algorithm(page 17, algorithm 3).

6.      The No-U-Turn sampler (NUTS) and Metropolis-adjusted Langevin algorithm (MALA) are suddenly introduced in Section 3.1, without prior mention or any references, and are then used for comparing the results obtained with pHMC. Given their relevance to the results section, introducing these MCMC variants earlier in the Background would enhance the paper's cohesiveness.

We added MALA and NUTS (which are not used in the subsequent result section, rather (p)HMC) in response to a previous question about our work (would NUTS or other samplers do a better job than (p)HMC on the Gaussian shell problem?).

The key point is that NUTS, MALA or (p)HMC are very unlikely to transition across the gap between the two Gaussian shells, so the answer to the question is no. The important addition here is the use of the Replica Exchange algorithm (it could be used with either NUTS, HMC or MALA). We could also run this test with HMC and get the same isolated shell. We propose mentioning in the text and caption that HMC produces similar plots to NUTS and MALA and explain this more clearly. We could also produce the figure with (p)HMC and simply mention that MALA and NUTS produce the same single shell, not exploring the other half. Good results have been reported for NUTS in hydrology e.g. (Xu et al., 2023; Krapu & Borsuk, 2022).

We have added references for MALA and NUTS (line 85) in the background and where used in the Gaussian shell example. We have kept all of the results but improved the wording in the figure 4 caption to:

Posterior samples for the Gaussian shells example obtained by different algorithms alongside the target distribution. Top left (a) is NUTS, top right (b) is REpHMC, bottom left (c) is MALA and bottom right (d) is the target distribution. Because of the addition of Replica Exchange, REpHMC can sample across the entire distribution space. This is in contrast to the NUTS, MALA and HMC (not shown) samplers which cannot transition across the gap between the two shells (figure 4).

7.      In Figure 6, the produced prior predictive 95% pointwise confidence interval seems quite narrow, which is unexpected given the variability one would anticipate when sampling from a 13-dimensional (prior) uncertainty space. Additionally, the observed discharge should have been plotted for comparison, to evaluate how well it is bracketed by the prior uncertainty interval.

We plan to remove Figure 6 and revise Figure 5 to include both the prior predictive data (not just one sample "synthetic discharge") and the observed discharge (already shown).

These priors provide small information (they are weakly informative), which is common for ODEs. As a result, the interval based on these priors is narrower than those that provide more information (informative priors), which usually have high variance relative to the location parameter. We have changed the priors to the same priors as in the real-world data section so we can compare the plot in Figure 5 with the calibrated model in Figure 16, as requested by the reviewer in point 12. Figure 6 has been removed. We now plot the prior predictive mean discharge instead of just one realisation. (figure 6)

[Figure]

8.      The Deviance Information Criterion (DIC) and Widely Applicable Information Criterion (WAIC) are introduced at the end of Section 3.2 without any explanation or references. In Paragraph 430, the IAT number and Geweke diagnostics are also mentioned without reference. It is unclear if these are assumed to be general knowledge. The placement of these terms is somewhat non-intuitive as they are subsequently used throughout the results section.

We will add these definitions, possibly in an Appendix - we already feel the paper is quite long!

References, descriptions, and interpretations of the WAIC and DIC have been added (lines 512 to 517).

We introduced the Geweke diagnostics and the IAT to check for convergence earlier between (lines 347 to 349) and provided a rationale for this method, along with references.

9.      The report lacks a clear statement regarding the number of forward model runs that were evaluated or needed. Is the correct interpretation that 10*4000 runs were conducted, multiplied by 15 for each model?

It's a good point, thanks for mentioning this. We haven't expanded on this point properly at all in the paper - we will add a general estimate of the number of forward model runs (and additionally, adjoint/backwards runs to get the gradient, that are of a similar complexity to the forward model run) and explain where they take place in the algorithm.

The leapfrog integrator requires L function and gradient evaluations for each sample, where L is the number of leapfrog steps. Therefore, there are L gradient evaluations in the leapfrog step for S samples and L*S*Number of chains.

We have added:

In terms of the theoretical complexity, if $N$ is the number of posterior chains, $S$ the number of samples per chain and $L$ the number of leapfrog steps per sample, then there are on the order of $NSL$ likelihood and likelihood gradient evaluations for the algorithm to complete (line 615).

*10.    The results of the synthetic experiments from Sections 3.2.1 and 3.2.2, depicted in Figures 10 and 13, are confusing. The Model 4 with four buckets (M_4) seems to be well calibration with the data originating from a much simpler model. This raises the question of why the hydrographs of Model 4 aligns so closely with those generated by the 'true' model, which would not be expected.*

Model 4 contains a superset of the components in Model 2 - consequently, it can reproduce the dynamics of the data produced from one of the simpler models (Model 2, in this case). We set this up intentionally to demonstrate that the Bayes factor will penalize the more parametrically complex model. However, we did not expand on this point properly - we will revise the text to make this clearer.

We have added a new sentence at the end of Section 2.1 Conceptual Models (line 171).

Note that for $i > j$ a more complex model $M\_i$ contains a superset of the modelling components of a simpler model $M\_j$. Consequently after calibration of both models on a dataset produced by $M\_j$, $M\_i$ should be able to reproduce the data as well as $M\_j$, but at the cost of higher model complexity. This setup will be used in the results to show that the \gls{bf} can penalise the complex model $M\_i$, leading to the selection of $M\_j$.

*11.    What does the conclusion from Figure 10 mean? "Hence, BF penalizes models with more parameters." How does one conclude this?*

This is connected to our response to question 10. The model fit is the same, so the BF penalizes the model with more parameters (the expected result). We will revise this caption to be more precise.

The new caption is:

Plot of the mean discharge data generated from the posterior predictive distribution of each model for experiment two. It is difficult to choose one model by inspection as they all fit the data well. However, the BF implicitly penalises the unnecessarily complex models M3 and M4 and correctly selects M2 (page 29,  figure 10).

*12.    In Section 3.3, the authors compare the uncertainty bounds in Figure 16 with a prior-predicted hydrograph from Figure 5. However, the hydrograph in Figure 5 represents only a single random realization from the prior, which seems like an inappropriate comparison. It would be more informative to compare the Monte Carlo mean derived from the prior with the mean hydrograph obtained from the learned posterior. As it stands, Figures 5 and 16 do not seem to be compared on an equitable basis.*

We agree, so we propose to change Figure 5 as proposed in our answer to Question 7.

Changes made in response to question 7.

*13.    The results and discussion in the results section have not convincingly demonstrated the ability of the prior calibrated posterior predictive p-value to detect prior data conflicts, a capability that was highlighted in the abstract and introduction.*

We will drop this point from the abstract and introduction as we agree it could almost justify a paper on its own (we will leave it in as a diagnostic, but not as a core part of the paper).

We have removed the emphasis on this feature of the paper in the abstract and introduction (line 6).

*14.    Section 3.4.2, titled "Convergence of marginal likelihood", feels brief and incomplete, as if the discussion in unfinished.*

We will add extra details of the marginal likelihood stabilizing for increasingly fine discretisation of the thermodynamic integral (TI) and some extra discussion on the convergence.

We have extended this section and included a plot showing the convergence of the log marginal likelihood in the number of temperature steps in Figure 3 and Figure 18 (pages 19 & 41).

*15.    The Nash Sutcliffe efficiency (NSE) obtained for the selected model is 0.397, which is low for a model deemed to be calibrated. Typically, NSE values below 0.6 are considered 'low'.*

We mentioned around line 503 that the NSE shows that the model is better than the mean, but we agree that the value is still 'low' - we will mention this in this text.

We note that we have now improved the model fit, so our original response is no longer appropriate (page 40, Table 8).

On the broader point of model (in-)adequacy, we agree that if these results were shown in the context of a paper proposing new hydrological models (defined by the operator G), there would be room for improvement. We picked this 'HBV-like' model with an extendable number of buckets largely for simplicity. However, the key contribution of the paper is on an approach for the model selection problem via Bayes Factors and in that context we think the results make sense.

Our wider point (line 160) is that the community should consider differentiable models and PPL as a standard methodology when developing new modelling toolboxes. This will open up the practical range of questions about models from parametric calibration (currently common) to model selection (still rare using BFs) enabled by algorithms like the ones we propose in this paper.

We have improved the model performance by tuning the number of leapfrog steps for the algorithm to an optimal number for all the models. This has improved the NSE to 0.526 and KGE to 0.704 for the chosen model, and updated all of the resulting statistics and plots (line 586).

*16. The results section does not sufficiently demonstrate the efficacy of the model. The findings presented in Figures 14 and 15 lack in-depth discussion. Although convergence diagnostics for real-world data suggest 'good' outcomes, the presentation falls short of being persuasive. Furthermore, the methodology appears to struggle with definitively identifying the most likely model in real-data scenarios, as indicated by the results in Table 8.*

We will add some extra explanations on Figure 14 and 15.

Figs 14. and 15. are unrelated to the model efficacy and simply show the output of the parameter calibration.

The second point we have already addressed in response to Question 15.

Table 8 identifies model 2 as the most likely model in the real-data scenario under the assumptions we choose to make (the true data-generating model is unknown and by definition, not in the predefined set of models). This is why we show many results for when the data generating model is in the set to persuade the reader that the algorithm works.

As discussed in the original response, for the real world data the BF definitively identifies model 2 as the most likely model under the assumptions made (lines 593 to 600).

*17. It would be beneficial to include a visual comparison, such as hydrographs, of the calibrated models M2 and M4 against the real data to better illustrate their performance.*

We will include the hydrographs against real data.

Tuning the number of leapfrog steps in the algorithm slightly improves the model performance. However, the model's structure is still quite basic compared to the 'full' HBV model. Additionally, based on the BF, M2 is now the preferred model, while the DIC and WAIC cannot distinguish between M2 and M3. As this is one of the most significant revision asked for by the reviewer we include the new figure below (fig 16).

[Figure]

18.    Based on the results of Figure 16, the uncertainty band drawn from the posterior seems wide, even for small streamflow values, which does not give a good hint of adequate parameter estimation / model calibration.

Indeed, the bands are wide, suggesting model improvements, hybrid models or modern approaches such as neural ODEs may be necessary from a hydrological modelling perspective.

**Technical corrections**

1.    Review the formatting and positioning of Equations 1 and 2e.

We aligned the equals but we can equally align on the left.

We have tweaked equation 2 but stuck with the alignment on the equals sign.

*2.      Clarify the paragraph containing lines 105, 110, and 115 to eliminate repetitive information and streamline the content for better readability.*

We will rewrite this part removing the duplicate sentences on DREAM, and merge the sentences on HMC, and generally make it easier to read.

We have removed the duplicate sentences on DREAM, merged the sentences on HMC and made some other modifications for readability.

*3.      Check Equation 2b for a possible typographical error: it should state $(V_2)t =$ and $n=2$, rather than $(V_1)t = n>=2$.*

We will correct this.

We have reconsidered our original response - the equation for the first bucket changes if there are two or more buckets (it has an outflow into the next bucket), so it is correct.

*4.      In Equation 3, there appears to be a typo with $k_{2,1}$, it should likely read $k_{1,2}$.*

We will correct this.

This has been corrected.

*5.      The use of 'p' to denote both the number of uncertain parameters (dimensionality of \Theta) and the number of discrete time steps in sections 2.1 and 2.2.1 is confusing. Consider using distinct notation for these two different concepts.*

We will make the notation distinct.

We now denote q as the number of timesteps to output the solution (lines 219 to 226).

*6.      The clarity of the paragraph on lines 210-215 could be enhanced. Simplifying the text and focusing on the key points would help to make the paragraph more comprehensible.*

We will simplify the text.

*7.      Confused by the structure – not sure why section 2.4, titled "Preconditioned Hamiltonian Monte Carlo" is a standalone subchapter. It may be more logical to include it in Section 2.3 "Numerical Methods" along with other algorithm steps.*

Indeed, this should be a subsubsection 2.3.x as it is an extension on HMC algorithm with a special inner product structure.

We have merged the HMC and pHMC subsubsections into a subsubsection so if now fits structurally with the rest of the paper (lines 350 to 389).

8.      *Address the overall layout beginning from page 25 to ensure that the content is well-organized and visually accessible to readers.*

We agree it's not very smooth at the moment. We are currently letting the text and figures flow according to LaTeX's default rules - after the final text is approved we will tweak the layout and pin the ordering to get a better flow through this section.

We played around with this but struggled to get anything substantially better - we will leave this to the final typesetting process with the journal.

9.      *The caption for Figure 8. "Posterior distributions for model M2" may be missing content. Should it be "Prior and posterior distributions for model M2" to accurately reflect the content of the figure?*

We will make this change.

The figure caption has been updated (page 27, figure 8).

10.     *Revise the description of Figure 10 for precision; it should likely specify that the "mean discharge data was generated from the posterior predictive distribution of each model and plotted," assuming that is the intended meaning.*

*We will adjust this.*

*We have adjusted the figure caption to that suggested by the reviewer (page 29, figure 10).*

11.     *Reformulate the awkward phrasing in line 465 to correct the sentence. It should read "which implies the model can generate the data", removing the extra "is".*

We will make this change.

Removed extra is (line 539).

12.     *Correct the reference error in line 475: "The mean log marginal likelihood is presented in Table 3" should be updated to "The mean log marginal likelihood is presented in Table 5" to direct readers to the correct table.*

We will make this change.

The table reference has been fixed (line 549).

"Selecting a conceptual hydrological model using Bayes' factors computed with Replica Exchange Hamiltonian Monte Carlo and Thermodynamic Integration" by Damian N. Mingo, Remko Nijzink, Christophe Ley, and Jack S. Hale.

**Reviewer #2**

https://doi.org/10.5194/egusphere-2023-2865-RC2

We would like to thank the second reviewer for their thoughtful comments. We will address their specific comments in this response and move towards a final response in the coming weeks. We are also more than happy to discuss specific points with the reviewer again.

*I have now finished reviewing the work by Mingo et al. The authors have combined Replica-Exchange Hamiltonian Monte Carlo (HMC) with Thermodynamic Integration in order to do Bayesian inference for the parameters of a conceptual hydrologic model, while simultaneously they compute the marginal likelihood of the model; the latter, facilitates model inter-comparison via the Bayes Factor (BF). In general, the manuscript is well written and has novelty in the sense that the proposed algorithm has never been applied before to hydrological modeling. As a result, I am overall positive! However, I think the manuscript would benefit from a more in-depth discussion (possibly toward the end of the article) about the scientific problem that the authors address, the limitations, and what are some possible alternatives.*

Indeed, there is an implicit assumption in our paper that computing the BF is something one might want to do in the first place! We agree we should have been more expansive on this point, so we make some specific answers to your points below. We will then paraphrase this into some new paragraph(s) in the discussion.

*In light of the extensive comments (major and editorial) of Reviewer #1 with which I completely agree, I would like to raise some concerns about the usefulness of BF as a hydrologic model inter-comparison metric. Please see my comments below:*

*1. For the synthetic experiments, Tables 4 and 5 show that both DIC and WAIC could correctly indicate the data-generating model, i.e., M2 and M3, respectively. For the average reader, this might practically mean that we do not need BF as an additional metric to "tell" us which model to choose. Please provide an explanation to show why employing BF matters. If you cannot demonstrate that the BF can capture the true underlying model while the other, simpler metrics, cannot, then it is hard to justify your analysis.*

There is an example of the BF succeeding to identify the underlying data generating model in our paper, whereas the DIC does not, and the WAIC only provides at best weak evidence.

In Experiment 2 (data generated from the three-bucket model M3), the DIC values for M3 and M4 differ by ~1, while the WAIC values for M3 and M4 differ by ~3 (Table 5 and Figure 12b).

(Burnham & Anderson 2002) on page 71, discusses ~4-7 (less evidence) and >~10 (substantial evidence) in favour of one model over another when using IC-type measures for model selection. There are other similar values in the literature but this seems to be a commonly used interpretation, akin to the table of Kass and Raftery for the BF (1995) which we show in Table 1.

Information and Likelihood Theory: A Basis for Model Selection and Inference. (2002). In K. P. Burnham & D. R. Anderson (Eds.), Model Selection and Multimodel Inference: A Practical Information-Theoretic Approach (pp. 49–97). Springer New York.
doi: 10.1007/978-0-387-22456-5 2

With this interpretation in mind, the DIC does not provide evidence to prefer M3 over M4. The WAIC possibly provides weak evidence in favour of M3 over M4, but we would be wary of making that conclusion by noting the substantial size of the error bar for M4 WAIC in Fig 12b. The BF (Table 5) decisively selects M3, the data generating model, over both M2 and M4.

This is clearly not evidence for the superiority of the BF as a model selection tool in all circumstances, and so we would not feel comfortable framing the BF as being superior in the paper. However, it is indicative that there may be cases where the BF succeeds where other approaches do not. The BF comes at a substantial computational cost over IC-type measures, necessitating improved algorithms such as the one we proposed in this paper if the BF is ever to be used at all in practice.

We will tweak the discussion of Experiment 2 to highlight this point better.

We have adjusted the discussion of Experiment 2 to reflect the discussion above as follows:

Following the recommendations in (Burnham and Anderson, 2002a) for interpreting information theoretic criteria, a difference of 4 to 7 suggests a weak preference for a model and a difference of at least 10 suggests strong preference for a model. Consequently, the DIC and the WAIC do not suggest a strong preference for the true model (M3) over the richer model M4. The WAIC shows possible weak evidence in favour of M3 over M4, but we note that the error bar in Fig. 12 for WAIC M4 indicates substantial uncertainty in the estimate. In this case then the BF decisively selects the data generating model M3 where the information theoretic criteria fail to do so. This example alone is clearly not proof that the BF is always superior to WAIC or DIC, but it suggests that there are cases in which BF succeeds and information theoretic criteria can fail. The success of the BF of course comes with a significantly higher computational cost (from line 552).

*2. Although I am not a Hydrologist myself, I have a hard time understanding the usefulness of BF within the context of hydrologic model comparison. Traditional hydrologists calibrate models*

*using algorithms like Shuffled Complex Evolution (SCE) based on optimization of a deterministic metric, e.g., NSE. I do understand that Bayesian inference of hydrologic model parameters, on the other hand, is appealing because it naturally provides a measure of uncertainty, which is always important. But the BF provides no pragmatic information to the modeler as per which model is performing better. For example, one would still have to compute NSE or KGE for all models M2, M3, and M4 for the real-world data (Table 8) to get an idea of what's happening. On the contrary, I would argue that for conceptual hydrologic models, which are not computationally demanding and time-intensive, likelihood-free methods like Approximate Bayesian Computing (ABC) might be more suitable for model comparison, as the posterior distributions of parameters for different models are obtained on the basis of an actually useful (to the modeler) distance metric, e.g., NSE, KGE, or even a metric tailored only to river discharge peaks!!!*

*Again, I am positive about your article and I believe it should be considered for publication, but please provide a better discussion about the practical use of BF as a hydrologic model comparison metric…*

We will first discuss the issue of the metric or goodness-of-fit measure.

In the paper we use a iid Gaussian likelihood function which in a deterministic setting can be seen as being equivalent to weighted minimisation in an $l^2$ norm, with the Bayesian prior being equivalent to some regularization term added to the $l^2$ norm.

According to (Cheng et al. 2014) the NSE is "equivalent to a log-likelihood function with iid Gaussian residuals". Consequently, if the modeler wishes to use NSE as a metric for parameter calibration (Cheng et al. 2014) proposes that they could simply use a iid Gaussian as a likelihood in a formal Bayesian analysis.

Qin-Bo Cheng, Xi Chen, Chong-Yu Xu, Christian Reinhardt-Imjela, Achim Schulte, Improvement and comparison of likelihood functions for model calibration and parameter uncertainty analysis within a Markov chain Monte Carlo scheme, Journal of Hydrology, Volume 519, Part B, 2014, Pages 2202-2214. https://doi.org/10.1016/j.jhydrol.2014.10.008

We are unable to find results formally linking the KGE with a likelihood function, which means that if the modeler wants to use KGE, they cannot use a formal Bayesian analysis. We found the paper (Liu et al. 2022) which derives an object called an 'informal pseudo probability density based on the KGE" which is then used in a formal Bayesian analysis, along with some discussion in the introduction of similar adaptations to the NSE. Perhaps this could then be used to evaluate the BF but this is a conjecture at this stage.

Yan Liu, Jaime Fernández-Ortega, Matías Mudarra, and Andreas Hartmann, Pitfalls and a feasible solution for using KGE as an informal likelihood function in MCMC methods: DREAM(ZS) as an example, Hydrology and Earth System Sciences,, 26, 5341–5355, https://doi.org/10.5194/hess-26-5341-2022

Likelihood-free methods such as Approximate Bayesian Computation (ABC) that bypass the evaluation of a likelihood function are a potentially good alternative if an explicit link with between a metric and the likelihood function is unavailable, as in the case of the KGE or perhaps with the example you mention with calibrating to capture peak discharge.

In summary, it seems that the adaptation of commonly used metrics such as NSE and KGE to a full Bayesian setting is still an active area of research. We will add a remark discussing metric choice and ABC to the section "Likelihood construction 2.2.1". This more detailed discussion will remain here if the reader is interested.

We now discuss the point on metrics vs model selection criteria.

The Gaussian likelihood (implying the weighted $l^2$ norm), NSE and KGE can be used as measures of goodness-of-fit for both training/fitting/calibration and nested model comparison via e.g. likelihood ratios. However, this is distinct from the model selection criteria (the BF or IC-type measures) which attempt to give a measure of fit balanced by an explicit or implicit penalisation for model complexity. Fit alone cannot choose between two models with free parameters which reproduce the data 'similarly' well. This is shown in our paper where we have deliberately constructed M4 with a strict superset of the model elements of M3 (and similarly M3 with M2).

So we agree that computing fit measures and performing graphical and formal posterior predictive checks are still an essential part of the modeling process. What the IC-type and BF offer is an additional measure for comparing models, potentially allowing the choice between models with similar fits.

We have added a paragraph at the end of the likelihood discussion:

We remark that according to (Cheng et al., 2014) our choice of a likelihood function with Gaussian white noise is equivalent to using the well-known Nash Sutcliffe efficiency (NSE) as a metric. However, other popular metrics such as Kling Gupta efficiency (KGE) cannot be linked explicitly with a likelihood function, and consequently cannot be used within a formal Bayesian analysis. A recent work (Liu et al., 2022) proposes an adaptation of the KGE idea using a Gamma distribution which can be used as an informal likelihood function within a Bayesian analysis, but we do not explore this option further here. An alternative option which bypasses the need for an explicit likelihood function is approximate Bayesian computation (ABC) could be an appropriate alternative when an appropriate explicit metric or likelihood function are unavailable see e.g. (Nott et al., 2012; Liu et al., 2023). (lines 227 to 234)

**Final response**

We would like to thank both reviewers again for their significant efforts towards reviewing the paper. We have made a number of small additional changes in response to the optional suggestions made by the second reviewer and some comments made by the external examiners at the first author's recent PhD defense. We reply inline below in green.

Review of the final manuscript: "Selecting a conceptual hydrological  model using Bayes' factors computed with Replica Exchange  Hamiltonian Monte Carlo and Thermodynamic Integration" by Damian N.  Mingo, Remko Nijzink, Christophe Ley, and Jack S. Hale1

Suggestions:

1. The Contribution section feels a bit too long, particularly the paragraph starting at line 110, which seems somewhat out of place and could potentially be moved…

We have decided not to adjust the contribution section.

2. Consider repositioning Figure 1; I believe it is generally preferred for figures to appear after they are referenced in the text.

Figure 1 can be repositioned in the final version, as it keeps moving.

3. In Equation 7, the meaning of 'D' is unclear—while it seems to represent a general discrepancy measure, it may be helpful to clarify this by adding a phrase such as, 'To compute the PPP, the chosen discrepancy measure, D, is calculated...'

Done in line 214.

4. It might be worth considering whether the method section, starting from 2.3, should be placed in a separate section.

We have decided not to split the sections.

5. I would suggest moving Table 3, which currently appears on page 22, closer to its first reference on page 19, as it is relevant to the data presented in Figure 6.

We have moved Table 3 closer to its first reference.

6. There is some confusion regarding the use of capital 'N,' which sometimes refers to the number of chains or replicas (e.g., Figure 2 and Algorithm 1) and at other times (e.g., page 21, line 445) to the number of posterior samples.

The N has been changed to S in lines 459, 530, and 531.

7. I noticed the absence of a concrete BF value, which is central to the paper. While it is mentioned that one can easily compute BF by just subtracting the values of the presented log marginal likelihoods (paragraph 455), I believe it would be beneficial to explicitly include a row in

the results tables presenting the BF values, as these are among the most important results of the paper.

The BF for a small number of model pairs has been added to the tables 4 and 5 (the others can be back calculated from the log marginal likelihood).

8. Insection3.3.1, the hydrographs presented in Figure16 are analyzed, particularly the two peaks that the model(s) failed to capture, corresponding to the dates 04/05- 02-1980 and 19/20-03-1980. However, it seems to me that the first peak not captured by the model's uncertainty band is actually closer to 20-02-1980, which to me, suggests a mismatch between the dates mentioned in the text and the data shown in the figure... though, it might be that I am wrong.

There was indeed a typo in the discussion and has been corrected - the second peak occurs on 19-02-1980 not 19/20-03-1980 (line 568). The date 19-02-1980 now matches the graph and the data. The conclusions do not change as this does not affect the results.

9. I found the last sentence in the conclusion regarding future work somewhat confusing. The sentence suggests combining REMC with NUTS to automatically turn all parameters in the HMC, but to me, it seems like this idea was not previously mentioned in the text as a possibility in any form…

We've improved this last sentence.